# Adaptive stratified importance sampling: hybridization of extrapolation and importance sampling Monte Carlo methods for estimation of wind turbine extreme loads

Peter Graf[1], Katherine Dykes[1], Rick Damiani[1], Jason Jonkman[1], and Paul Veers[1]

National Renewable Energy Laboratory, Golden, CO, 80401 - USA

*Correspondence to:* Peter Graf (peter.graf@nrel.gov)

**Abstract.** Wind turbine extreme loads estimation is especially difficult because turbulent inflow drives nonlinear turbine physics and control strategies, so there can be huge differences in turbine response to essentially equivalent environmental conditions. The two main current approaches, extrapolation and Monte Carlo sampling, are both unsatisfying: extrapolation-based methods are dangerous because by definition they make predictions outside the range of available data, but Monte Carlo methods converge too slowly to routinely reach the desired 50-year return period estimates. Thus a search for a better method is warranted. Here we introduce an adaptive stratified importance sampling approach that allows for treating the choice of environmental conditions at which to run simulations as a stochastic optimization problem that minimizes the variance of unbiased estimates of extreme loads. Furthermore, the framework, built on the traditional "bin"-based approach used in extrapolation methods, provides a close connection between sampling and extrapolation, and thus allows the solution of the stochastic optimization (i.e., the optimal distribution of simulations in different wind speed bins) to guide and recalibrate the extrapolation. Results show that indeed this is a promising approach, as the variance of both the Monte Carlo and extrapolation estimates are reduced quickly by the adaptive procedure. We conclude, however, that due to the extreme response variability of turbine loads to the same environmental conditions, our method and any similar method quickly reaches its fundamental limits, and that therefore our efforts going forward are best spent elucidating the underlying causes of the response variability.

## 1 Introduction

Estimating extreme loads for wind turbines is made especially difficult by the nonlinear nature of the wind turbine physics combined with the stochastic nature of the wind resources driving the system. Extreme loads, such as those experienced when a strong gust passes through the rotor or when a turbine has to shut down for a grid emergency, can drive the design of the machine in terms of the material needed to withstand the events. The material requirements in turn drive wind turbine costs and overall wind plant cost of energy. Thus, accurate modeling and simulation of extreme loads is crucial in the wind turbine design process. This paper discusses the use of adaptive importance sampling in estimation of such loads. Importance sampling (IS) (Robert and Casella, 2004) is a well established method for using samples from one distribution to estimate statistics from another. Adaptivity in importance sampling has been introduced in (Karamchandani et al., 1989) and elsewhere, but does not appear to have broadly taken hold. Here we introduce an adaptive importance sampling method for extreme loads estimation.

The essential task in wind turbine extreme loads estimation is to evaluate the probability of exceedance (POE) integral

$$P(Y > l) = \int P(Y > l|x) f(x) dx. \tag{1}$$

Here $P(Y > l)$ is the probability of load $Y$ exceeding target/threshold $l$, $P(Y > l|x)$ is the conditional probability of exceedance given wind speed $x$, and $f(x)$ is the distribution of wind speeds (or other environmental conditions). Because we are interested in extremely low probability events and $Y(x)$ is stochastic and only available via simulation, standard methods of integration do not apply.

Existing approaches fall generally into two main classes. The first is based on extrapolation: data is gathered in different wind speed "bins", extreme value distributions are fit to the empirical distribution function for each bin, and these are then integrated. The second is based on Monte Carlo methods: exceedance probabilities are written as expectations of indicator functions, samples are drawn from an assumed wind distribution, and unbiased estimates made by the usual Monte Carlo summation. Unfortunately, to date neither of these approaches is satisfactory. The crux of the difficulty is that on the one hand too many samples are required for converged *Monte Carlo* estimates, but on the other hand reliable *extrapolation* of nonlinear physics under uncertain forcing is extremely problematic, especially without knowledge of the form (e.g. quadratic, etc.) of the nonlinearity. Nevertheless, the computational expense of MC implies that except in rare cases, some sort of extrapolation will be necessary in order to reach the desired 50 year return period estimates. This paper is motivated by the intuition that perhaps we can at least use MC/IS to make sure extrapolations are accurate to the resolution of data we actually have, and to gather data in ways that accelerates their convergence.

The difficulty of estimating these "tail-probabilities" of interest in extreme loads estimation is one of *timescales*. We are trying to estimate loads seen roughly once in 50 *years* using a set of simulations whose total length is only a few *hours*. This large difference in timescales means that any uncertainty in the data is necessarily magnified by the extrapolation. Small variations in short-term data could lead to significant over- or underestimation of long-term extreme loads.

One of our main conclusions will be that while we may have reasonable knowledge of the distribution of environmental conditions a turbine faces, we have very little knowledge regarding the distribution of the *response of the turbine* to its environment, and this response variability may in fact be so large that our knowledge of the distribution of environmental conditions is of limited use. A conceptual aid is provided by the Inverse First Order Reliability Method (IFORM) (Winterstein et al., 1993; Sultania and Manuel, 2017) and a variant of it called the Environmental Contour (EC) method, which will be discussed briefly below. These methods, though highly practical in their own right, are not the main subject of the present paper, but they will help us appreciate the important distinction between environment and response.

Our goal is to develop methods that make unbiased estimates that *minimize variance* as a function of the number of samples/simulations, and to use these to dynamically update extrapolations. Our proposed method, Adaptive Stratified Importance Sampling (ASIS), is essentially a global stochastic optimization method where the search variables are the number of samples from each wind speed bin we use, and the objective function is the variance of our Monte Carlo estimates. The key tool here is importance sampling, which allows us to continually produce unbiased estimates of exceedance probabilities even as the dis-

tribution of bins changes. These quasi-optimal samples are then used to make the best possible extrapolations. Results below show that this is indeed a promising approach.

The organization of this paper is as follows. First we present the necessary background on the existing extrapolation method (e.g., as recommended in the IEC standard (IEC, 2005)), Monte Carlo methods, and importance sampling. Next we describe our ASIS algorithm. Then we present a brief study illustrating some of its properties. The paper provides a context for discussing extrapolation and importance sampling in the same framework. Our conclusion highlights the potential for this approach, reiterates some of the fundamental difficulties with the endeavor, and leads to suggestions for where we should next focus our efforts to crack this difficult problem.

## 2 Background on turbine simulation, extrapolation, Monte Carlo, and importance sampling

### 2.1 Turbine simulation

Throughout this paper, we use FAST, NREL's aeroelastic simulation tool (Jonkman, 2013). FAST is a widely-used industry and academic tool for wind turbine loads estimation. NREL's WISDEM software allows for the execution of FAST and its companion tool TurbSim (which generates turbulent wind fields for input to FAST) in a programmatic fashion from `python`, as has been reported previously (Graf et al., 2016).

The particular turbine on which we are testing these methods is the NREL 5 MW reference turbine, often used for such studies (Jonkman et al., 2015; Choe et al., 2016), in an onshore configuration. The "environmental conditions" are thus described by hub height mean wind speed (modeled by a Weibull distribution with scale and shape parameters of 11.28 m/s and 2, respectively). Additional environmental parameters of turbulence intensity, spectrum, coherence function, and wind shear are kept fixed at nominal values. It should be noted that the stochasticity in the combined TurbSim/FAST simulation comes from the "random seed" that is input to TurbSim. This seed governs the exact starting conditions for the generation of the turbulent inflow wind field. Because this is an ergodic system, a long enough simulation would eventually cover all possible wind conditions. However, for finite simulations, multiple runs using different random seeds allow us to "sample" the space of all possible turbulent flow field snapshots with the same mean wind speed, turbulence intensity, etc. This is common practice in extreme loads analysis. We can regard random seed as a proxy for sampling over a uniform distribution of turbulent inflows for each set of environmental conditions.

For this study, we selected two output channels of interest, tower base side-side bending ("TwrBsMxt" in FAST nomenclature) and tower base fore-aft bending ("TwrBsMyt"), which provide contrast because the wind speeds where their highest loads occur overlap differently with the typical wind speed distribution. The side-side moments grow with hub-height wind speed, making their extremes hard to estimate with traditional MC sampling because they do not overlap well with typical wind distributions. The fore-aft counterparts do overlap quite closely with the typical wind distributions.

## 2.2 Extrapolation

The current standard for estimating extreme loads relies on extrapolation. We refer the reader to the relevant literature for a detailed exposition of the extrapolation method (Ragan and Manuel, 2008; Moriarty, 2008; Toft et al., 2011; Graf et al., 2017). Here we present a concise statement of the method and discuss one or two subtleties. The protocol to construct exceedance curves using binning and extrapolation is as follows:

1. Run Turbsim/FAST $N_i$ times per wind speed $x_i$ at center of bin $i$ (Typically $N_i \sim 6$).

2. For each bin $i$, concatenate the data from each seed and extract peaks (see below r.e. peak extraction and time scales). For future reference we refer to the resulting dataset as $\{Y_{i,k}\}$ where $i$ indexes over wind speed bins and $k$ indexes over the peaks we have extracted at that wind speed

3. For each bin $i$, form empirical cumulative distribution functions (CDFs) and fit a chosen distribution $F_i(x) = P(Y < l|x)$ to them. In this paper we use a 3 parameter Weibull distribution.

4. (optional) For each bin: convert each fitted distribution to desired time scale (see below r.e. peak extraction and time scales).

5. Finally, $P(Y < l) = \int P(Y < l|x)f(x)dx \sim \sum_i P(Y < l|x_i)f(x_i)\Delta x_i$, where $f(x)$ is pdf of wind distribution, and $\Delta x_i$ is the width of bin $i$. The extrapolated estimate of the probability of exceedance is then $P(Y > l) = 1 - P(Y < l)$.

The distributions chosen to fit the bin-wise CDFs are the theoretically appropriate extreme value distributions (Generalized Extreme Value (GEV), 3 parameter Weibull, etc.). However, this does not mean they accurately represent the behavior of the particular FAST loads in a specific context. The optimality properties of extreme value distributions are asymptotic properties, but we are doing "intermediate asymptotics": long term–but not infinitely long term–trends. Nevertheless, these distributions are the appropriate starting point.

In this paper we are using a 3 parameter Weibull distribution, but this is not meant as a claim that this choice is better than any other in the literature (Gumbel, GEV, etc). We used the 3 parameter Weibull because we have used it with success in previous work (Graf et al., 2017). There are many excellent studies examining the choice of distribution (Ragan and Manuel, 2008; Moriarty, 2008; Toft et al., 2011), but this is not the focus of the present work.

Regarding the fitting procedure, in light of the interest in extrapolation, rather than just fitting, we have fit the empirical cumulative distribution function of the data directly to the theoretical cumulative distribution function (CDF) of the distribution by nonlinear least squares. We have done this separately for the data from each wind speed bin. Furthermore, in order to emphasize the largest peaks (i.e. the lowest probability values) we do not use all the data, just the $M_{pks}$ largest peaks in each bin, where $M_{pks}$ is an algorithmic parameter. $M_{pks}$ plays a role in our studies similar to the "threshold" used in peak-over-threshold methods, but for purposes of connecting to the bin-based approach it has the advantage that there are always the same number of peaks extracted from a given length simulation. As an exercise, we experimented with using different values of $M_{pks}$, as discussed below in section 4 and illustrated in Figure 2. The broad conclusion is that there is a window of values

of this parameter that provides similar performance, which suggests this parameter will not be an impediment to practical implementation of this algorithm.

It is important to be clear regarding various time spans at play here. First, there is the ultimate time of interest, typically in wind studies the "50 year return period". This does not mean that in 50 years the event in question happens with probability 1. Sometimes it is loosely defined as an event having probability $1/50$ of happening in one year. Strictly speaking it is an event that happens on average one or more times in 50 years according to a Poisson process whose "events per interval" parameter is one in 50 years. Next there is the simulation time, i.e., the length of each FAST run. This is almost always 10 minutes in the literature. Relatedly, there is the time span over which our estimates of exceedance probability apply. These are also traditionally 10 minutes, i.e. reported probabilities of exceedance are probabilities of exceedance *in 10 minutes*, but there is nothing in principle to make this fixed. Finally, there is the length of time between independent peaks. This time can be estimated empirically by examining the autocorrelation of the data; values as low as 4 seconds have been justified in previous studies (Ragan and Manuel, 2008), and 10 seconds seems to be more than adequate.

The various time spans come into play as we extract peaks and make estimates. The rule that connects them is the simple "AND" rule of probability: If $Y < l$ for time $T$, and $T = Kt$, then $Y < l$ for time $t$ for $K$ times in a row, so $P_T(Y < l) = P_t(Y < l)^K$. For probabilities of exceedance $P(Y > l)$, we write $P(Y > l) = 1 - P(Y < l)$ and use the same idea, resulting in the familiar expression $P_T(Y > l) = 1 - (1 - P_t(Y > l))^K$. For example, the former is typically used to convert peak distributions collected on a 10 second, 1 minute, or peak-over-threshold basis to a 10 minute basis, while the latter is used when we derive the ubiquitous value of 3.8 x $10^{-7}$ that represents the probability of the 50 year return period event happening in 10 minutes.

In this paper we adopt the following setup: All our simulations are 11 minutes long, where we discard the first minute as a transient and retain the final 10 minutes for our studies (we will occasionally be loose with the terminology and refer to these as "10 minute simulations"). To gather peaks, we take the maximum of each 1 minute segment in our simulations. This provides exactly 10 peaks per simulation, which allows for building 1-minute empirical cumulative distribution functions (i.e. probability of exceedance in 1 minute) in a consistent manner. The resulting 1 minute empirical POEs are converted to 10 minute POEs as described above (i.e. $K = 10$) to conform to standard practice. Below we experiment with how many of these peaks should be used in the fitting of the extreme value distributions used in the extrapolation method. This alternative to the peak-over-threshold method makes comparison and correspondence between extrapolation and sampling easier, because we have a fixed number of peaks per FAST run. The tradeoff between gathering more peaks (at the risk of sacrificing statistical independence) versus less peaks (at the risk of not having enough data) is an algorithmic detail that we could study further, but it is not the focus of this paper. Our motivation for using 1 minute interval peak separations is to pick a reasonable point along this tradeoff that avoids the pitfalls of either extreme. Finally, we have divided the wind speed range into 5 bins centered at 8, 12, 16, 20, and 24 m/s.

To acquire a sense of the basic variability of the response, we have run 20 independent sequences of the simulations described above for the extrapolation method (6 random seeds per bin). Figure 1 consists of box and whiskers plots of the peaks from the first 6000 peaks (10 peaks per run X 6 seeds X 20 repetitions X 5 bins). For both the side-side and fore-aft loads, the variability

within each bin is large. For example, the difference between 95th and 5th quantile for the 24 m/s side-side bin is comparable to its median value. For the side-side load the dependence on wind speed is clearly also very strong, whereas for the fore-aft load, the variability *within* the bins is as large as it is *between* the bins.

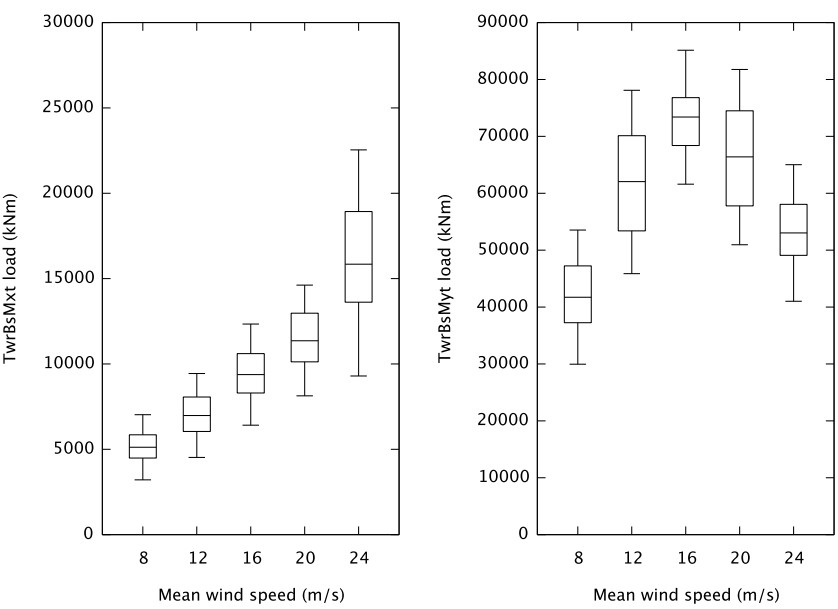

**Figure 1.** Box and whiskers plots of the distribution of raw response (specifically, all 1-minute maxima) of the combined TurbSim/FAST simulation as a function of wind speed bin for side-side (left) and fore-aft (right) loads. The difference in general trend between side-side and fore-aft loads is clearly evident. The variability within each bin is extreme (esp. for the fore-aft load), which puts an upper bound on the utility of sampling methods (e.g., importance sampling) targeting certain wind speeds. The boxes show the median, 25th, and 75th percentile. The whiskers are positioned at the 5th and 95th percentile. The data are the absolute maxima in 1 minute segments of 120 separate 10 minute simulations per bin (1200 total peaks for each bin).

### 2.3 Monte Carlo importance sampling for extreme loads

5   Monte Carlo (MC) methods are widely used to estimate expectations of quantities calculated by stochastic simulations (Robert and Casella, 2004). Importance Sampling (IS) is an MC method in which an auxiliary *importance distribution* $q(x)$ is used to focus sampling on areas of the target distribution $f(x)$ that are most relevant with respect to the functions of interest (e.g., the loads $Y(x)$). Often "relevant" means *minimal variance* (see below). The broad applicability of the method arises from the

so-called Importance Sampling Identity:

$$E_f[Y(x)] = \int Y(x)f(x)dx \tag{2}$$

$$= \int Y(x)\frac{f(x)}{q(x)}q(x)dx \tag{3}$$

$$= E_q[Y(x)\frac{f(x)}{q(x)}] \tag{4}$$

where $E_f$ and $E_q$ represent the expectation with respect to $f$ and $q$, respectively. This means, from an MC standpoint, that both

$$E_f[Y(x)] \sim \frac{1}{M_{tot}} \sum_i^{M_{tot}} Y(x_i) \qquad \text{with } x_i \text{ drawn from } f, \tag{5}$$

and

$$E_q[Y(x)\frac{f(x_i)}{q(x_i)}] \sim \frac{1}{M_{tot}} \sum_i^{M_{tot}} Y(x_i)\frac{f(x_i)}{q(x_i)} \qquad \text{with } x_i \text{ drawn from } q, \tag{6}$$

(where $M_{tot}$ is the total number of samples we have of the quantity we are estimating) are *unbiased* estimates of the same quantity $E_f[Y(x)]$.

For us $M_{tot}$ will be the number of peaks gathered from the FAST runs. In what follows we will use $M$ to represent numbers of peaks, and $N$ to represent numbers of FAST runs. With our convention of taking the maximum over 1 minute spans of 10 minute simulations, we will have $M = 10N$ throughout. The subscript "$_{tot}$" will be the total number (over all bins), whereas index subscripts (e.g. "$_i$") will refers to the peaks or runs within the corresponding bin. Thus as above $N_i$ is the number of FAST runs (i.e. random seeds) in the $i$th bin, and $M_i = 10N_i$ is the number of peaks extracted from the $N_i$ runs, and $N_{tot}$ is the total number of FAST runs.

Although the above estimates (5) and (6) are both unbiased, they could have drastically different *variance*, which means that they may converge at drastically different rates. The minimal variance importance distribution can be derived and is

$$q^*(x) = \frac{Y(x)f(x)}{E_f[Y(x)]} \tag{7}$$

From a practical standpoint there are two obvious problems with this result. First, it depends on the expectation we wanted to calculate in the first place. This objection we can overcome using some form of accept-reject sampling that does not depend on the normalization constant. [As long as we can evaluate $Y(x)$, even if by simulation (we assume we can also evaluate $f(x)$), we can sample from any distribution proportional to $Y(x)f(x)$ by the accept-reject algorithm (see, e.g., (Robert and Casella, 2004)), which involves sampling uniformly in a 2D region containing the function $Y(x)f(x)$. The probability of $x$ w.r.t. the $Y * f$ distribution is just the proportion of these uniform samples below $Y * f$ in this 2D "box". This procedure does require assumptions on the bounds of $Y$ and the support of $Y$ and $f$, but in principle these can be made large enough to sample any reasonable probability $Y * f$]. A more significant problem, however, is that $q^*$ also depends on the function $Y(x)$ whose expectation we are trying to calculate with as few evaluations as possible. Finally, in our case, an even worse problem is that

$Y(x)$ is not a deterministic function of $x$, so, as stated, $q^*$ is also a stochastic function. Nevertheless, Eqn. (7) provides a guide for our quest to find the minimal variance importance distribution: it should be as close as possible to proportional to the product of load $Y(x)$ and wind probability $f(x)$.

For our purposes, finally, note that Eqn. (1) can be written as an expectation of the so-called "indicator" function that is 1 if $Y(x) > l$ and 0 otherwise:

$$P(Y > l) = E_f[I(Y > l)] \sim \frac{1}{M_{tot}} \sum_i I(Y(x_i) > l) \frac{f(x_i)}{q(x_i)} \qquad \text{with } x_i \text{ drawn from } q. \tag{8}$$

Eqns. (1) and (8) form the basis of the mathematical bridge between extrapolation and MC/IS methods described in section 3.

## 2.4 IFORM

Here, keeping in mind the goal–minimal variance unbiased estimates of extreme loads through importance sampling, minimizing the use of extrapolation–we summarize the Inverse First Order Reliability Method (IFORM) and Environmental Contour (EC) methods. IFORM was introduced by Winterstein (Winterstein et al., 1993) and addresses the estimation of extreme loads from a different perspective. Instead of directly computing the integral in Eqn. (1), IFORMS seeks to find the combined set of environmental conditions and resulting loads that have a desired joint return period probability. The EC variant explicitly separates the environment from the turbine's response to it, in effect being a method that seeks to directly find the conditions that *cause* the extreme load.

In the general IFORM approach, the combined environmental and response variable space is considered as one joint probability distribution, and the quantile corresponding to the desired return period is explored to find the maximal response. In practice, the distribution of the environmental part of this combined space is assumed known. For example, in this paper, the environmental component is wind speed, which is assumed to have a Weibull distribution with shape and scale 11.28 m/s and 2, respectively. Then response data is gathered for samples in the environmental space, and a conditional distribution "response | environment" is *fit* to this data, which allows finally for *extrapolation* to the desired return period. This use case of IFORM is thus a form of extrapolation, with its strengths (it requires very few samples to build arbitrarily low POE estimates) and weaknesses (there is no guarantee, as outlined above, that extrapolation outside the range of data, i.e., from short to long times, is valid). In this sense everything we can say about using adaptive importance sampling to reduce the variance of traditional extrapolations applies to IFORM estimates as well. The more data we have *in the relevant bins*, the more accurate our statistical fits and resulting extrapolations will be.

The EC variant of IFORM explicitly separates environment from response. It works best if the response of interest is a completely deterministic function of environmental conditions that themselves have known probability. Then one can directly search the environmental contour (e.g. all wind/wave/turbulence/etc. combinations that occur on average once in 50 years) to find the highest load. Otherwise, a conditional distribution of response subject to environment can model the response variability away from its median; in this case, EC is then similar to IFORM in practice. Together, IFORM and EC solidify the important notion of *response variability*: the magnitude of the variation of the nonlinear stochastic response for *fixed* environmental conditions.

These notions help to explain why IFORM or EC applied to wind turbine extreme loads estimation may not be much different than other extrapolation methods. On the one hand, as noted above, though systematic and efficient, IFORM relies on extrapolation. Just like the standard extrapolation method, it relies on being able to extrapolate from easily observable quantiles (5th,25th,50th,75th,95th,...) to the very difficult to observe quantiles corresponding to the 50 year return period. On the other hand, EC is not applicable when the main driver of variation in a system is the response variability, which (see Figure 1) is largely the case for wind turbines; the main driver of extreme loads is not a particular wind speed but some idiosyncratic chaotic process that even for a prosaic wind speed just might come together to cause an extreme load.

The present task, that of estimating extreme loads with wind speed as the only environmental variable, is governed mostly by the response variation. Therefore we will not estimate extreme loads by IFORM in this paper. However, IFORM is critical for conceptual understanding: where possible, our goal should be to "convert" response variation (intractable) to environmental variation (tractable) through better understanding of its physical cause. It is the extreme response variability (different random seeds for the same environmental conditions can cause very different FAST output, because they cause very different turbulent inflow) that makes extreme loads estimation a difficult problem. We return to this subject in section 5.

## 3 Adaptive stratified importance sampling (ASIS)

### 3.1 Bin-wise empirical CDFs as the bridge between extrapolation and MC

When we perform the bin-wise simulations used in the extrapolation methods, we are performing *stratified* sampling. Recognizing that these samples can be described as a probability distribution provides a bridge to using them in an IS context, as discussed in (Graf et al., 2017). The basic idea is as follows. Assume a set of samples $\{x_i\}$ from an arbitrary distribution $g(x)$ of wind speeds ($g$ may be either $f(x)$, $q(x)$, or an empirical one derived from binning the data). By running FAST, the set $\{Y_i\}$ of corresponding loads can be generated and sorted from the lowest to the highest. Then for any given load $Y_j$, the IS estimate is:

$$
\begin{aligned}
P(Y < Y_j) \quad &= \quad E_f[I(Y < Y_j)] & (9)\\
&= \quad \int I(Y(x) < Y_j)f(x)dx & (10)\\
&= \quad \int I(Y(x) < Y_j)\frac{f(x)}{g(x)}g(x)dx & (11)\\
&\sim \quad \frac{1}{M_{tot}}\sum_i I(Y(x_i) < Y_j)\frac{f(x_i)}{g(x_i)} \quad \text{with } x_i \text{ drawn from } g & (12)\\
&= \quad \frac{1}{M_{tot}}\sum_i \begin{cases} \frac{f(x_i)}{g(x_i)} & \text{if } i < j \\ 0 & \text{otherwise} \end{cases} & (13)\\
&= \quad \frac{1}{M_{tot}}\sum_{i<j} \frac{f(x_i)}{g(x_i)}. & (14)
\end{aligned}
$$

Letting $w_i \equiv \frac{f(x_i)}{g(x_i)}$, we can concisely write the empirical CDF for *all* the loads $\{Y_j\}$ as:

$$P(Y < Y_j) = \frac{1}{M_{tot}} \sum_{i<j} w_i. \tag{15}$$

To apply this formula to data from the binning method, we need the appropriate $g(x)$, i.e. the appropriate weights $w_i$ of each sample. Recall, we write the dataset as $\{Y_{i,k}\}$ where $i$ indexes over wind speed bins and $k$ indexes over peaks extracted at that

wind speed. We rewrite the integral over wind speeds as a sum of integrals, and then approximate each separate integral by the $M_i$ peaks derived from the $N_i$ runs (in our case, over TurbSim random seeds) at the fixed wind speed $x_i$ for bin $i$:

$$\int I(Y(x) < Y_j) f(x) dx \quad = \quad \sum_{i=1}^{N_{bins}} \int_{x_i}^{x_{i+1}} I(Y(x) < Y_j) f(x_i) dx \tag{16}$$

$$\sim \quad \sum_{i=1}^{N_{bins}} \frac{1}{M_i} \sum_{k=1}^{M_i} \begin{cases} f(x_i)\Delta x_i & \text{if } Y_{i,k} < Y_j \\ 0 & \text{otherwise} \end{cases} \tag{17}$$

From this, we see that the "weight" contributed by sample $i, k$ to the probability of non-exceedance (1-POE) of $Y_j$ is $\frac{1}{M_i} f(x_i)\Delta x_i$

for all $i, k$ s.t. $Y_{i,k} < Y_j$. To calculate the empirical CDF from the bin data, then, we assign weight $w_i = \frac{1}{M_i} f(x_i)\Delta x_i$ to all samples from bin $i$ and apply Equation (15). Table 1 summarizes the exact correspondence between extrapolation and IS/MC. The importance density corresponding to stratified sampling is seen to be $g(x_i) = \frac{M_i}{M_{tot}} \frac{1}{\Delta x_i} = \frac{N_i}{N_{tot}} \frac{1}{\Delta x_i}$.

**Table 1.** Bin-wise empirical cumulative distribution functions provide a bridge from extrapolation, which builds POE from bin-wise fitted distributions $F_i$, and importance sampling, which builds an unbiased estimate from the appropriately defined distribution $g$.

| Method | $P(Y < l) = \int P(Y < l \mid x) f(x) dx$ | Remarks |
|---|---|---|
| Empirical bin-wise CDF | $\sum_i \frac{1}{N_i} \sum_k I(Y_{i,k} < l) f(x_i)\Delta x_i$ | $P(Y < l \mid x_i) \sim \frac{1}{N_i} \sum_k I(Y_{i,k} < l)$ |
| Extrapolation | $\sum F_i(l) f(x_i)\Delta x_i$ | $P(Y < l \mid x_i) \sim F_i(l)$, fitted to above |
| Importance Sampling | $\frac{1}{M_{tot}} \sum_{i,k} I(Y_{i,k} < l) \frac{f(x_i)}{g(x_i)}$ | sampling from $g(x_i) = \frac{N_i}{N_{tot}} \frac{1}{\Delta x_i}$ |

Thus we have a "bridge" between fitting and sampling. Bin-wise empirical CDFs can be directly compared with fitted distributions (in fact, they are what we fit to). But the estimate over all wind speeds can then be expressed generically in the

IS language suited to comparison with MC and IS estimates. IS/MC do not provide any "bin-wise" information (there are no bins), so it is otherwise impossible to "debug" their divergence from extrapolation. This formulation allows us to see, first, that error accrues from lack of convergence of empirical CDFs, for both methods. Additionally though, for extrapolation, the error is compounded by lack of fit between the chosen extreme value distribution and the empirical CDFs, which is the price we pay for being able to extrapolate to arbitrarily low POEs with small numbers of samples.

## 3.2 ASIS as stochastic optimization

The discussion above indicates that the samples from the bin based methods can alternatively be used to make empirical estimates via their implied importance distributions. This orientation suggests, also, that there is no barrier to changing the distribution of samples as we go. So we can think of the estimation procedure as an optimization problem: find the distribution of bins (number of samples per bin) that results in unbiased estimates with minimal variance. In (Graf et al., 2017) we have used a heuristic algorithm that looked for "gaps" in the empirical peak distribution. Here instead we introduce a gradient-based approach. As above, let $N_i$ be the number of FAST runs performed in the $i$th wind speed bin, and let $N$ be the *vector* of bin-counts. Now, the variance of the estimate using importance distribution $g$ is

$$J(N) \equiv var_g[P(Y > l)] = E_g[\frac{I(Y > l)^2 f(x)^2}{g(x)^2}] - E_g[\frac{I(Y > l)f(x)}{g(x)}]^2. \tag{18}$$

Note the second term does not depend on $g$ (in the corresponding integral, the $g$ in the denominator cancels out). So

$$\frac{\partial J}{\partial N_j} = \frac{\partial}{\partial N_j} E_g[\frac{I(Y > l)^2 f(x)^2}{g(x)^2}] \tag{19}$$

$$\sim \frac{\partial}{\partial N_j} \sum_{i,k} I(Y_{i,k} > l)^2 f(x_i)^2 \frac{N_{tot}^2 \Delta x^2}{N_i^2} \tag{20}$$

$$= -2 \sum_k I(Y_{j,k} > l)^2 f(x_j)^2 \frac{N_{tot}^2 \Delta x^2}{N_j^3}. \tag{21}$$

Here we have used the fact that $\frac{M_i}{M_{tot}} = \frac{N_i}{N_{tot}}$ to write the expression in terms of bin counts instead of peak counts.

Our algorithm begins by running the standard 6 seeds per bin from the extrapolation method (i.e., $N_i$ is initialzed to 6 for all $i$). Then we perform the following steps in an iterative fashion:

1. Compute $\nabla_N J(N)$.

2. Allocate a target number of new samples (e.g. 20 per iteration) to bins in two ways:

    (a) allocate some percentage of the new samples in proportion to $\nabla_N J$,

    (b) recognizing this is a *global* optimization problem, allocate the rest to other bins randomly.

3. Run TurbSim/FAST for the new batch.

4. Update our empirical estimates of POEs *and* our extrapolation estimates.

Note the algorithm as stated does not explicitly recognize the stochasticity of the underlying quantity $Y(x)$. Because we are using an unbiased estimate of the gradient of the variance, our approach, naive as it is, is known to converge "almost surely" to a local optimum (Robbins and Monro, 1951). In fact it is a form of Stochastic Gradient Descent (Goodfellow et al., 2016). Casting the problem in this form allows for taking advantage of ongoing research in this area. There are two reasons for step 2 (b) of the algorithm. First, it is not clear a priori that our optimization problem is convex. Thus there could be multiple

local minima. Steps 2 (a) and 2 (b) correspond to the tradeoff between "exploitation" and "exploration" common to all global optimization algorithms. Second, because the gradient is calculated from an unconverged statistical estimate, there is an error associated with this vector. Preventing unconverged estimates from steering us in the wrong direction is an additional reason to include the "explore" step 2 (b).

5     A slight complication comes from the need for the $N_i$ to be integers (we can only have an integer number of runs per wind speed bin). Currently we simply round $N_i$ to the nearest integer. Thus we are following the gradient as closely as possible subject to the integral nature of $N_i$. We claim that this is a reasonable procedure, because the variance of the estimates is observed to be a rather slowly varying function of the bin distribution, so taking the nearest feasible (i.e. integral $N_i$) point to the point suggested by our gradient will incur a bounded and likely small error. Also, the gradient itself is an estimate, not an 10 exact value, so we are already working with inexact quantities from the standpoint of traditional gradient-based optimization. A more refined approach we could explore in the future is to apportion simulation *time* spent in each bin according to $N_i$. This would amount to a "relaxation" (from the discrete space to the continuous space (Parker and Rardin, 1988)) of the problem and would allow for exactly following the (albeit still stochastic) gradient.

    Next, as stated $J$ only concerns one load type (e.g. tower base side-side bending moment). But previously (Graf et al., 2017) 15 and above (i.e. Fig.1) we have seen that side-side and fore-aft loads favor different importance distributions. Since it defeats the purpose of the method to have to repeat it for every load, we have adopted an "umbrella" concept; we compute the desired bin distribution for all the loads of interest, and form the minimal superset of bins that includes them all. Also, note that the gradient (e.g., see Equation (21)) is always negative; increasing *any* bin count will reduce the variance, which makes obvious sense. The purpose of the algorithm is to guide the *distribution* of bin counts to have optimal proportions from each bin. Our current 20 implementation works in a cumulative fashion. At every iterations, we are always adding more simulations, never removing them.

    Another issue, even for a single load type, is how many of its peaks $Y_i$ are considered in computing the gradient of the variance. We remind the reader that ASIS per se does not require any extrapolation. This question of how many peaks to use to compute $\nabla_N J(N)$ is different from the question of how many peaks $M_{pks}$ to use for extrapolation. It is another algorithmic 25 parameter that one could tune. Using all the peaks would reduce the variance of the POE estimates of all the peaks, even the small peaks we don't care about. Using just the single largest peak would overemphasize the bin that this single peak happened to come from. The choice of the 5 largest peaks is rather arbitrary, enough so that more than the single largest peak contributes, but not so many as to deemphasize the goal of finding *large* peaks.

    Finally, because we are still refining the mechanics of the algorithm, ASIS as stated does not include a stopping criteria. 30 Since what ASIS minimizes is the variance of our load estimates, stopping should be based on driving the variance below a user-defined threshold. The difficulty, of course, is that unlike a deterministic gradient descent procedure, our only access to the actual variance is through further statistical estimates. In the results below we simply repeat the stochastic optimization procedure 100 times and compute the variance of the estimated loads directly. A less computationally expensive approach is *bootstratpping* (Fogle et al., 2008; Sultania and Manuel, 2017), which we would recommend for a production implementation 35 of ASIS. In bootstrapping, the variance of our load estimates are estimated as follows: At every iteration, there are $M_i$ peaks

extracted from the $N_i$ runs at wind speed $x_i$ for bin $i$. Normally we use these directly in Equation (15) to form a single POE estimate. Instead, in bootstrapping, we resample the $M_i$ peaks *with replacement* some number, say $L$, times to form $L$ resampled sets of peaks, all of length $M_i$, but all containing different subsets of the original $M_i$ peaks. These are used to compute $L$ independent POE estimates, from which an empirical variance can be computed and used for stopping criteria.

## 4  Results

In this section we demonstrate the basic mechanics of the algorithm in the context of a study of the effect of the number of peaks, $M_{pks}$, used to fit the bin-wise extrapolation distributions ($F_i \sim P(Y < l | x_i)$, above). First, Figure 2 illustrates the variability of the extrapolated exceedance probability as a function of $M_{pks}$. We have run FAST for 10 minutes 20 separate times for each bin (the figure shows only the results for the 20 m/s bin, but the others are similar). This results in 200 minutes of total simulation time, thus according to our 1-peak-per-minute convention (which is fixed throughout the paper), 2000 peaks. Each line on the figure (e.g. "5 peaks", "10 peaks", etc.) is the result of fitting the 3 parameter Weibull CDF to just the largest $M_{pks}$ (e.g. $M_{pks}$ = 5, 10, etc) of these 2000. The line labelled "empiricial CDF" is the empirical CDF of the 2000 peaks. It is important that each peak always represents the same amount of simulation time. Otherwise we would have to re-weight the contribution of each peak to the POE in Equation (15).

As an exercise, we examine the sensitivity of the ASIS results to $M_{pks}$. For this, we study the variance of the resulting extrapolation, which we can estimate simply by repeating the entire sampling, simulating, fitting, and extrapolation procedure 100 times. (Note, to estimate the variance in a production environment where compute time was of paramount importance we would recommend the more efficient if less straightforward bootstrapping procedure described above). The results are summarized in Figures 3, 4, and 5.

For each of the 100 independent tests, we ran extrapolation and ASIS for 25 iterations (an iteration of extrapolation is simply re-performing the extrapolation procedure with the current ASIS bin data), which adds a varying number of new samples to each bin at each iteration. The most obvious observation is that indeed the variance of the estimates is decreasing quickly as a function of iteration. ASIS reliably drives the variance of the estimates of POE down, and simply recalculating the extrapolations to "keep up" with ASIS drives the variance of the extrapolation estimates down as well. There is a slight dependence on $M_{pks}$, but it does appear there is a "sweet spot" around 40 peaks that is good for both loads (further study of the optimal $M_{pks}$ is beyond the scope of this paper; it is akin to the study of the optimal threshold in the peak-over-threshold method). The standard deviation drops by roughly a factor of 3 after only about 100 FAST runs, (compared to 30 for the original extrapolation (iteration 0)). This is closer to a "$\frac{1}{N}$" rate of convergence than the theoretical "$\frac{1}{\sqrt{N}}$" convergence of Monte Carlo integration.

Thus our "adaptive extrapolation" approach appears capable of reducing variance somewhat dramatically with minimal additional computation. The two approaches maintain correspondence even while adapting the bin distribution, which allows for leveraging the variance reduction of the empirical ASIS estimate to reduce the variance of the extrapolation estimate. And the latter is the estimate of real importance, because that is what will be used in practice. Note that ASIS could as well drive

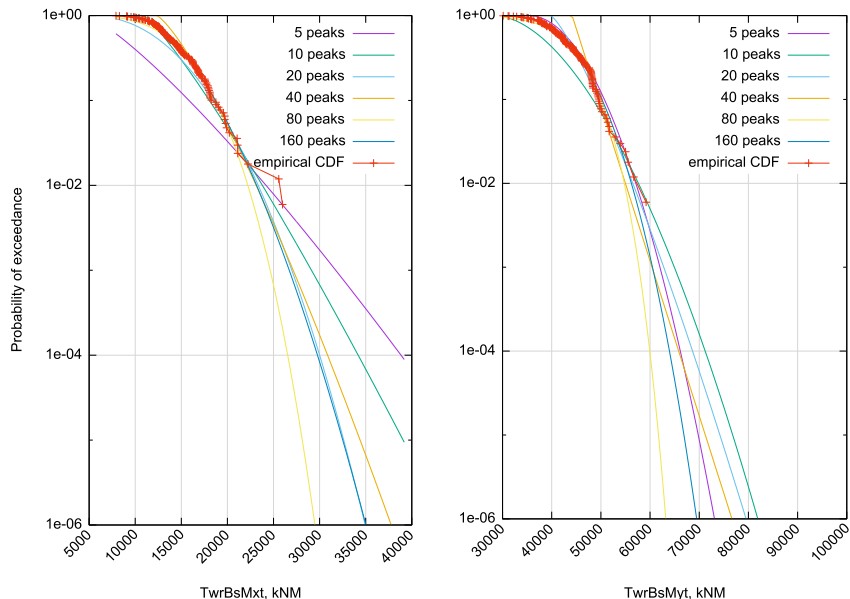

**Figure 2.** Weibull 3 parameter fits to empirical CDF for wind speed bin centered at 20 m/s for a variety of choices for how many peaks $M_{pks}$ we use for the fit for side-side (left) and fore-aft (right) tower base loads. In all cases we are fitting the analytical CDF to the *highest* $M_{pks}$ loads of the empirical CDF (a lower values of $M_{pks}$ is like a higher threshold in the peak-over-threshold method). Especially for the side-side moment, the extrapolated POE values depend heavily on how many peaks are used.

IFORM estimates instead of the traditional extrapolation estimates. In both cases ASIS optimizes the distribution of samples (wind speed bins in this case) that are then used to fit statistical distributions, which are then used to extrapolate to desired return periods.

## 5    Conclusions

5    In this paper we have built a bridge between bin-based extrapolative methods and sample-based importance sampling Monte Carlo methods. With this, we proposed an adaptive stratified importance sampling (ASIS) algorithm that is both more efficient than existing Monte Carlo approaches and maintains contact with the extrapolation methods and thereby allows for iteratively increasing the extrapolation accuracy. This is important, because only the extrapolations are able to routinely make estimates of extremely long return period load exceedance probabilities.

10    The search for the optimal importance distribution is a stochastic optimization problem. As stated above, our algorithm is a convergent algorithm. But stochastic optimization is an active area of research, and more sophisticated algorithms may exist to improve our approach. We need to keep in mind, however, that the optimization problem is a means to an end. The real goal is minimal variance estimates with the smallest amount of effort. We want to *use* the optimal importance distribution at the

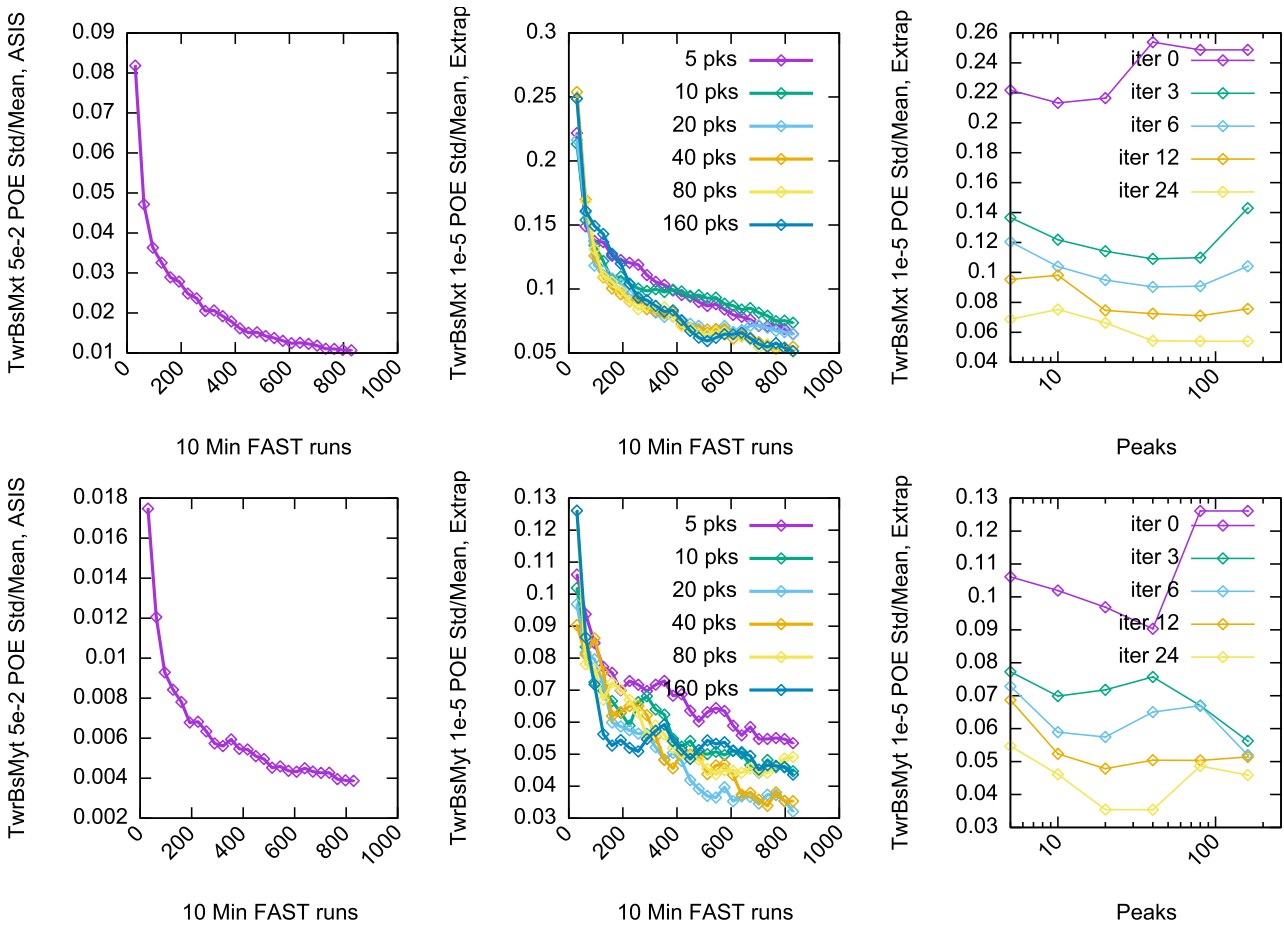

**Figure 3.** Summarizing the convergence of tower base load estimates using extrapolation and ASIS estimates over 100 independent runs. The top row is side-side, the bottom is fore-aft. The y-axis units are the ratio of the standard deviation to the mean estimate (relative standard deviation, measuring convergence). The x-axis is the number of FAST runs (measuring computational expense). The target POE for the empirical ASIS estimate (left) is $5 \times 10^{-2}$ and for the extrapolation estimate (middle and right) is $10^{-5}$. Adaptively selecting samples in a way designed to accelerate the convergence of the empirical estimate (ASIS, left) also accelerates the convergence of the extrapolation estimates (middle, right). There is a somewhat weak dependence on the number of peaks used for extrapolation, but $M_{pks} \sim 40$ appears robust for both loads. The side-side load estimate has larger initial relative variance, because (as shown above in Fig. 1) its extremes occur at high winds, but its relative variance is reduced more quickly by the adaptive procedure than the fore-aft load.

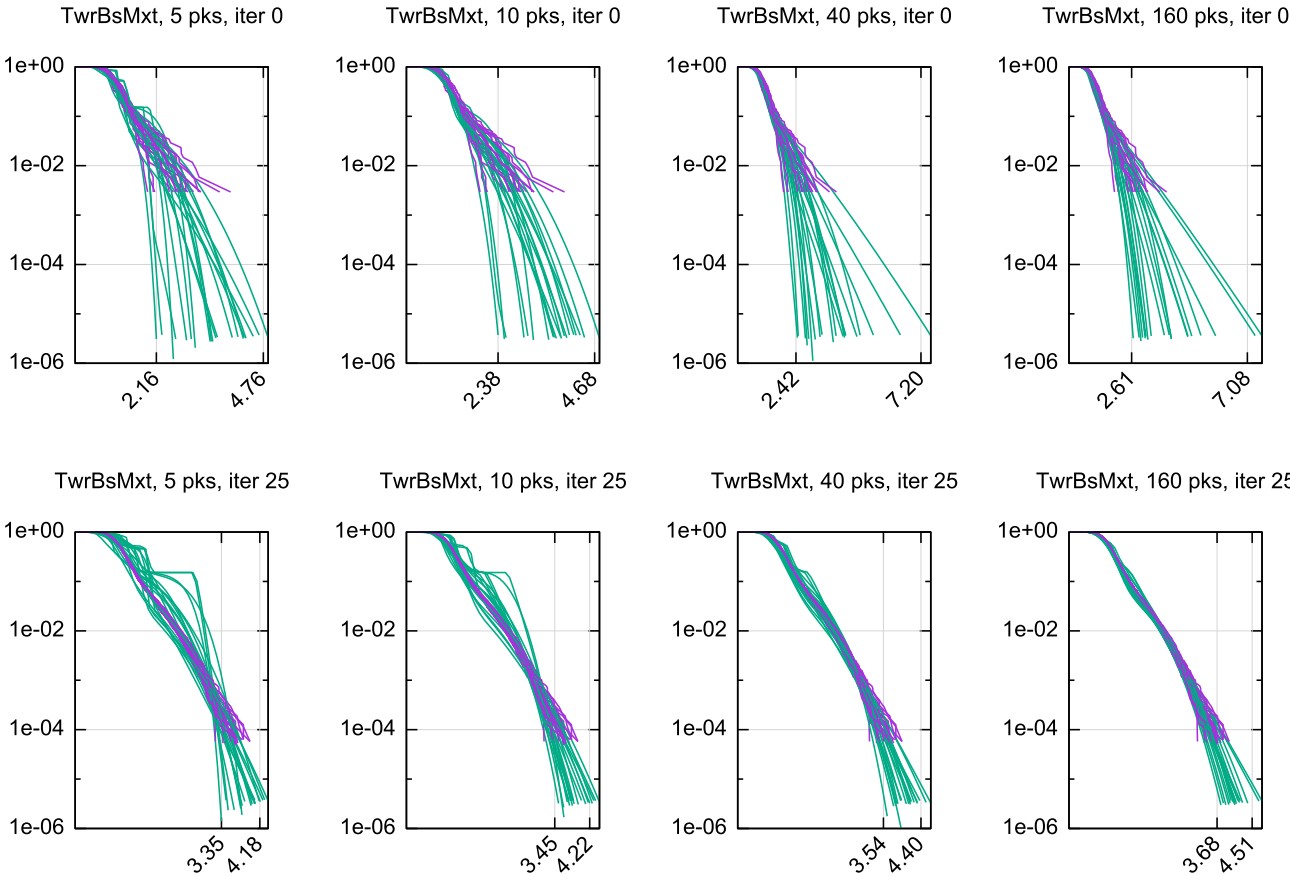

**Figure 4.** Behavior of ASIS and iterative updating of extrapolation for side-side tower base bending load over 20 separate runs as a function of $M_{pks}$. The x-axis unit are 10,000s of kN-m, the y-axis is probability of exceedance in 10 minutes. The top row shows the results of both ASIS and extrapolation at iteration 0 (i.e. just based on the initial set of bin-wise samples) as a function of the number of peaks used for fitting the extrapolation distributions. The bottom row shows the estimates after 25 ASIS iterations. (Note the ASIS results are the same across each row because they are independent of $M_{pks}$.) Clearly the variance of the estimates is tightened. It is not clear from visual inspection if one choice of $M_{pks}$ is better than any other.

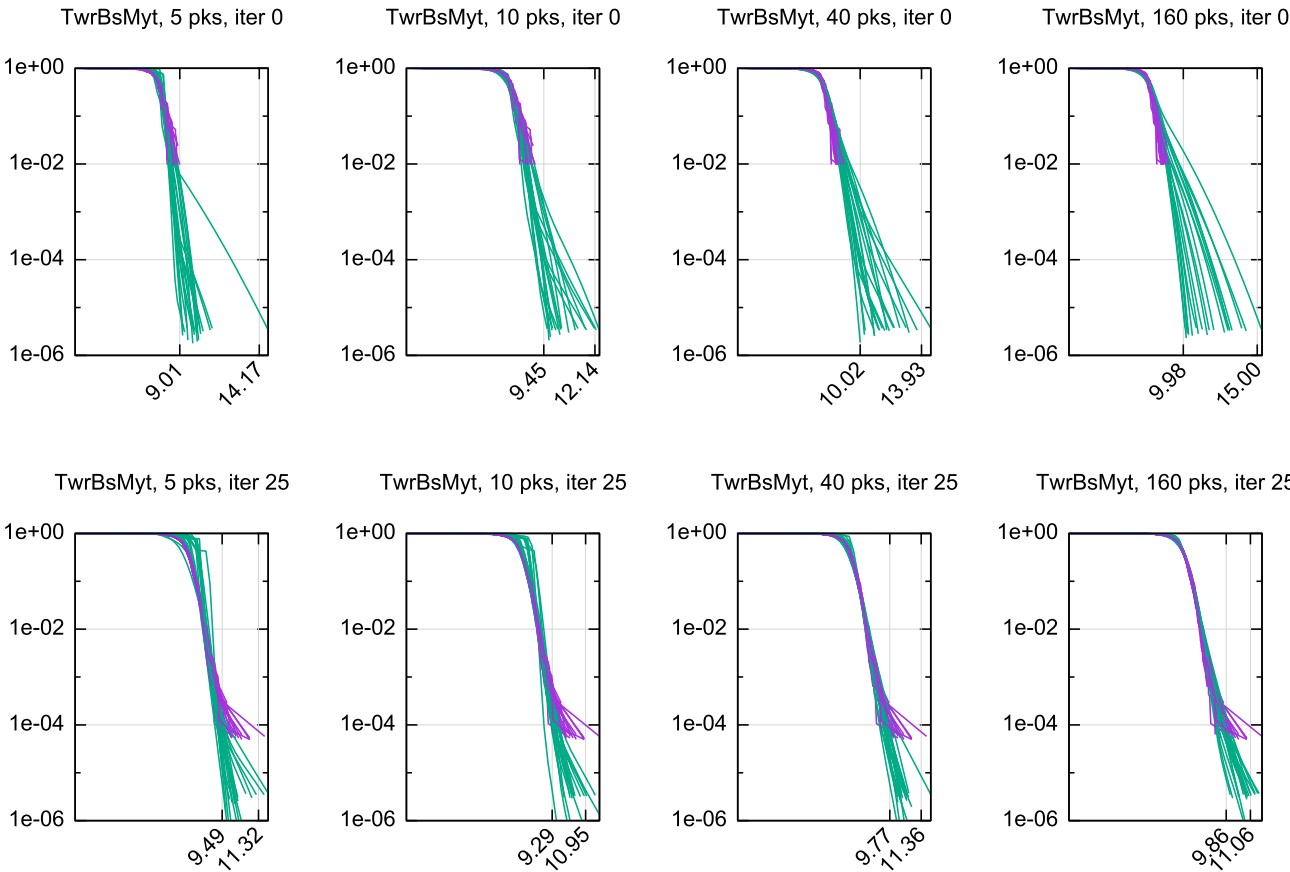

**Figure 5.** Behavior of ASIS and iterative updating of extrapolation for fore-aft tower base load over 20 separate runs. The x-axis unit are 10,000s of kN-m, the y-axis is probability of exceedance in 10 minutes. The format is the same as Fig. 4. Again it is interesting to compare the visual representation with the statistics presented in Fig. 3. The single extremely large load discovered by ASIS was also seen in (Graf et al., 2017) (Fig 5, panel (f)). Though it is beyond the scope of the present paper to do so, one of our main conclusions is that the statistical methods have come to a point where the best course forward will be to pursue the exact causes of such loads and integrate a statistical description of such situations into our methods.

same time as we are discovering it. Relatedly, we need to also keep in mind that we have the dual mission of both efficiently estimating the load POEs *and* accurately estimating their *variance*. We can use the peaks we sample to make unbiased estimates of variance just as we do expectation, but these are only *estimates*, and they themselves suffer from lack of convergence. The resampling method of bootstrapping described above offers a way to leverage a single data set to estimate statistics *and their variance*, and in a practical setting this would be recommended (as opposed to the completely separate runs we have described above).

In principle there is no barrier to application of ASIS to higher dimensional problems. In particular, it is well known that turbulence intensity and turbulence standard deviation have a large role in wind turbine extreme loads (Bos et al., 2015), where a bimodal importance distribution is warranted. It would be interested to see if the variance minimization of ASIS would discover this distribution. This would open the door to trusting an automated procedure to derive the distribution.

This problem may be ripe for a machine learning approach: The physics *is* in the solver. To the extent it is possible, we should be able to learn from increasing amounts of data. For this, we need accurate variance estimation methods that can build a "loss function" for learning algorithms that examine data and decide how to process it to make the best next estimate, and to choose the best next places to sample; here we have presented a framework for extrapolating from such data that allows for learning the best extrapolation strategy from the variance minimization algorithm.

On the other hand, we should realize there *is* a *physical* source of extreme response variation, which is the combination of turbulent inflow and nonlinear turbine response. By "opening up the black box", i.e., circling back to the original physics, we hope to *transfer* what in the present setup is response variability into the realm of environmental variability, at which point we can use its probability distribution to hone in on just the loads of interest (i.e., the extreme loads) more quickly. Further studies into the root causes of extreme response variation of wind turbine loads and their ultimate incorporation into more efficient statistical extreme loads estimation are ongoing.

*Acknowledgements.* The authors would like to express their gratitude to professor Lance Manual for conversations on this topic, especially his insights into the IFORM and EC methods, as well as his extremely thoughtful reading of the manuscript.

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
