# Peer review of "Adaptive stratified importance sampling: hybridization of extrapolation and importance sampling Monte Carlo methods for estimation of wind turbine extreme loads"

_Wind Energy Science, 2017_

## Referee Comment (RC1) · L. Manuel (Referee) · 26 Aug 2017

*Discussion of:*

**Graf, P. et al. "Adaptive stratified importance sampling: hybridization of extrapolation and importance sampling Monte Carlo methods for estimation of wind turbine extreme loads" [DOI:10.5194/wes-2017-30]**

Lance Manuel

*lmanuel@mail.utexas.edu*

August 26, 2017

**Overview and General Comments**

This is a most interesting presentation of a vexing problem that has proven to be a challenge to wind turbine loads analysts for many years. The ideas developed by the authors and the narrative discussing the desire to "bridge" more conventional extrapolation methods and variance reduction techniques that go beyond brute-force Monte Carlo simulations are welcome. Casting the problem as an optimization problem, albeit without the usual formalisms, so as to adaptively improve estimates of long-term loads is done most effectively. Throughout, there are interesting insights and discussions that make for an illuminating reading and exposure to the essential issues.

**Specific Section-by-Section Comments**

**Introduction**

The description about IFORM on Page 2, Lines 12–13 should be clarified. More correctly, only with the environmental contour (EC) method which is the most commonly employed version of IFORM, one uncouples the "environment" from the "response," and global extremes of interest associated with a target return period are approximated by using the maximum response from among all response levels derived only for candidate environmental variables consistent with that return period (in other words, response variability given environmental conditions is neglected). That said, in cases such as the one described in this article, where the environment is described using only one random variable (wind speed), the EC method has limited use. The EC method is better suited when a pair of random variables (say, wind speed and turbulence intensity, or wind speed and wave height for offshore turbines) are included. When only one random variable defines the environment, the "environmental contour" for a target return period is reduced to a single wind speed. This has limited use because, for instance, in the present example, for both side-to-side and fore-aft tower bending moments, the method will suggest that only rare and high wind speeds around cut-out that are associated with a 50-year return period need to be considered in turbine aeroelastic simulations. This will clearly lead to inaccurate 50-year fore-aft tower moments.

Now, despite the preceding comment that the EC method reduces the conventional environmental contour to a degenerate point or single-valued wind speed to consider for turbine response simulations, in fact, in the present case, one could instead use IFORM in its more general form and use wind speed and response as two random variables and formally derive estimates of 50-year side-to-side and fore-aft tower bending moments. This is discussed later, along with comments offered in the context of Section 2.4 (IFORM).

**Extrapolation**

Page 4, Line 14: Strictly speaking, the 50-year return period event is that event that is exceeded "on average" once in 50 years. Even though it is not the same in general, sometimes the event is defined as one that is exceeded "on average" with a probability of 1/50 in one year.

Page 4, Line 26: $3.8^{-7}$ should be $3.8 \times 10^{-7}$.

**Monte Carlo importance sampling for extreme loads**

Page 6, Line 23: The comment that some form of accept-reject sampling can be used with importance sampling is an intriguing one. It is unclear how exactly this would be done given that $Y(x)$ is not known in closed form; any additional notes, even if included very briefly, regarding such sampling would help.

**IFORM**

As stated earlier, the EC method doesn't apply here as there is *no* environmental contour corresponding to the authors' example—such a contour is a degenerate single wind speed value obtained as $F_V^{-1}(3.8 \times 10^{-7})$ and as such has limited value for, say, the fore-aft tower bending moment where the derived 50-year load will certainly be under-predicted. Indeed, in this single environmental random variable case, the degenerate single simulation needed for *any* response or load of interest would require simulations to be run for a single wind speed above cut-out, i.e., for $V$ equal to 43.3 m/s. This would be meaningless.

The authors correctly point to the deficiencies of IFORM (on Page 7) but, in light of comments in the preceding paragraph, since there is no environmental contour at all that can be defined to describe their example study, much of the extended discussion regarding the EC method and environmental contours as presented in Section 2.4 is not relevant.

Now, in a most interesting way, the very issue that doesn't allow for a critique of the EC method—namely, that the authors choose only $V$ as an environmental random variable—actually allows the more general IFORM procedure to be used with the authors' own simulation results and will lead to reasonable results (how this can be achieved is presented here very briefly). The idea is as follows: Consider that there are two random variables—wind speed, $V$, and the response or load, $Y$, whose statistics are derived from 10-min simulations . We will assume that we know the probability distribution for $V$ (for instance, here, the authors use a Weibull $V$ with shape and scale parameters equal to 2 and 11.28 m/s, respectively); we establish conditional distributions for $Y$ given $V$ based on simulations. We can use IFORM, though *not* the EC method, to find the required quantiles of $Y|V$ for any $V$ of interest. This is purely a geometry problem (involving mapping of $V$ and $Y$ to two independent standard normal random variables). To illustrate this, because the results presented in Figure 1 are the easiest to read off and learn from without great effort, one would find using IFORM that for TwrBsMxt, the 24 m/s bin would require that the desired 50-year response must have a probability of exceedance in 10 minutes of $5.95 \times 10^{-6}$. Given the 5th, 25th, 50th, 75th, and 95th percentile loads in Figure 1, a 2-parameter Weibull fit to these data leads to a 50-year TwrBsMxt value of 29,700 kN-m. Other (lower) wind speeds occur more often and associated load levels to be checked for those wind speeds using the IFORM procedure must be rarer, i.e., with exceedance probabilities in 10 minutes that are smaller than $5.95 \times 10^{-6}$. Given the data, these wind speeds do not lead to TwrBsMxt values at the desired probability levels that exceed what was found for $V$ equal to 24 m/s. In a similar manner, for TwrBsMyt, selecting $V$ equal to 16 m/s, the desired 50-year response for IFORM must have a probability of exceedance in 10 minutes of $7.20 \times 10^{-7}$. Again, from the data in Figure 1, a 2-parameter Weibull fit to these data read off easily, leads to a 50-year TwrBsMyt value of 94,300 kN-m. Again, note that other wind speeds and associated (different) response quantiles need to be checked, as part of the IFORM procedure, to ensure that the largest load quantile across all the wind speed bins is then claimed as the 50-year load. Details regarding all the calculations are not presented here but IFORM computations are based on the Weibull $V$ and loads data from Figure 1.

In sum, the authors' comment regarding searching on the environmental contour (or just inside it) is not pertinent here. There is also no need to discuss above-median response levels in this context. Both the preceding comments would have been appropriate if, in addition to wind speed, another environmental random variable were included such as turbulence intensity. As illustrated, IFORM in its general form can be easily employed here and response variability can be directly accounted for—as the authors state correctly, this variability is ignored by the EC method. It is *not* ignored by IFORM in general and, as shown in the previous paragraph, even with the limited data presented in Figure 1, the method works quite efficiently in deriving 50-year loads. By running

additional simulations at critical wind speeds, the resulting loads data and subsequent distribution fits to the same will lead to reduced uncertainty in derived 50-year loads.

**Adaptive stratified importance sampling (ASIS)**

Reference to POE, on Page 8, Line 20, should really be to the probability of non-exceedance.

In Step 2 of the algorithm on Page 10, why not simply obtain new samples (with rounding) in proportion to $\Delta_N J(N)$? It is not clear but it seems that the most important bin (where $\Delta_N J(N)$ is largest) is allocated some samples and the remaining (of 20) are then randomly allocated to other bins. Why randomly?

The algorithm, as presented, appears not to state what is the criterion for stopping or convergence. The discussion regarding the "umbrella" concept that suggests a minimal superset of sample distributions across bins is exactly what is needed. It is the only way to guarantee adequate samples of response extremes to meet very distinct response characteristics such as between TwrBsMxt and TxrBsMyt. The ASIS algorithm as presented doesn't explicitly state this but assumes convergence when all the response measures are adequately sampled in all bins so as to yield unbiased 50-year response values, presumably with some specified confidence level on these predictions. If it helps, in an offshore wind turbine application, Sultania and Manuel [3], employed bootstrap-based confidence intervals for specific sea states (akin to bins here) to arrive at the appropriate number of simulations for accuracy in response probabilty distributions, conditional on the environment. Given the results presented in this article, it appears that convergence on long-term loads for each bin is indeed achieved by the authors in their examples by examining the variance of loads associated with low POE levels.

It was not clear, upon reading, what was the reason for using the 5 largest loads. The largest loads will automatically drive the tail of long-term loads distributions and, as such, the 50-year load, when these largest loads are included along with *all* smaller loads; so, why retain only the 5 largest? Some clarification would help here.

**Results**

The results in Figures 3–5 are very interesting and suggest that the methodology proposed by the authors offers a robust and efficient means of deriving long-term turbine loads for design. The rapid reduction in variance in estimates of long-term loads by using the ASIS algorithm is convincing.

A few observations are offered.
1. It would be very useful and insightful to see how $g(x)$ or the bin-wise importance sampling changes with iteration for the the two load measures, TwrBsMxt and TwrBsMyt, separately, and what the ultimate umbrella sampling ends up being, after convergence, or as the number of iterations changes—given the contrasting characteristics of the two load measures, one might expect a bi-modal sampling with one mode around or slightly above rated and another closer to cut-out will result.
2. It is not completely clear what $M_{pks}$ refers to. For instance, does $M_{pks} = 40$ imply that, after accounting for samples from all bins, only the largest 40 are used or does it mean that for each bin, the largest 40 loads are retained and then combined with 40 from the other bins before extrapolation?
3. The non-monotonic upward trending cooefficient of variation on the 0.05 POE load seen in Figure 3 for a very large number of peaks might be caused by dependence among the peaks that results when too many peaks are extracted from each 10-min simulation (see Fogle et al [2]).
4. Is it possible that the $x$ axis units in Figure 4 are 10,000's of KN-m. rather than 1,000's of kN-m as stated? This would be consistent with Figures 1 and 2 which appear to show TwrBsMxt loads an order of magnitude higher. It would also be consistent with the IFORM-based estimates that were computed (above) as $29.7 \times 10^3$ kN-m and $94.3 \times 10^3$ kN-m, respectively, for TwrBsMxt and TwrBsMyt. The caption for Figure 5 is correct; the one for Figure 4 might need to be corrected.
5. As presented, Figures 4 and 5 do not show loads for the low POE level of $3.8 \times 10^{-7}$ associated with the target 50-year return period. It would have been useful to see those results. Indeed, ASIS-based convergence at the lowest POE levels that are presented, suggest that 25 iterations

and the use of 40 peaks is very good.

In the caption for Figure 3, $5^{-2}$ should be $5 \times 10^{-2}$.

**Conclusions**

The closing discussion is most helpful in setting this work in the context of other studies regarding the derivation of long-term loads for wind turbine design. The ASIS algorithm, as presented, will prove useful to loads analysts. While the authors have demonstrated its efectiveness using 100 independent tests in their numerical studies, in practice, it may be useful to suggest use of the ASIS algorithm followed by bootstrapping, after each iteration, and confirmation that the coefficient of variation on some load quantile (as in Figure 3, for example) is acceptably low—for example, 5%. It is clear that ASIS can achieve this target with a far smaller number of aeroleastic simulations than extrapolation based on ordinary or conventional sampling methods. The authors do, in fact, recognize in their closing discussions, that bootstrapping with a single data set could be used with ASIS and thus reduce computational effort.

Only to provide a contrast with the authors' work, it should be noted that IFORM-based approaches, referred to in this article, can be very efficient. This has been demonstrated in this discussion using the authors' own data (crudely derived from their Figure 1); note that in the illustration presented response variability is *not* ignored as is done with the environmental contour method. Accounting for response variability in IFORM is not difficult—it is conceivable that an ASIS-like formulation for sampling could prove far more efficient than even IFORM but this needs to be demonstrated for situations where more than one random variable defines the "environment."

Finally, related to this last point, the ASIS algorithm would be especially important to employ along with the introduction of stochastic turbulence and with the treatment of turbulence intensity or turbulence standard deviation explicitly as a random variable. The role of gusts and turbulence in extreme loads is known to be significant and the need then for bivariate importance sampling distributions could present challenges (Bos et al. [1], van Eijk et al. [4]); at the same time, the benefits to be derived from approaches such as ASIS, if its efficiency and accuracy is demonstrated in such cases, would then add to its appeal.

On Page 15, Line 3, "stratified adaptive" should be "adaptive stratified" to be consistent with the ASIS acronym.

**References**

[1] R. Bos, W. Bierbooms, and G. van Bussel. Importance sampling of severe wind gusts. *11th EAWE PhD Seminar on Wind Energy in Europe*, 2015.

[2] J. Fogle, P. Agarwal, and L. Manuel. Towards an improved understanding of statistical extrapolation for wind turbine extreme loads. *Wind Energy*, 11(6):613–635, 2008.

[3] A. Sultania and L. Manuel. Reliability analysis for a spar-supported floating offshore wind turbine. *Wind Engineering, (DOI: 10.1177/0309524X17723206)*, 2017.

[4] S. F. van Eijk, R. Bos, and W. A. A. M. Bierbooms. The risks of extreme load extrapolation. *Wind Energy Science*, 2(2):377–386, 2017.

---

## Referee Comment (RC2) · Anonymous Referee #2 · 28 Aug 2017

**General comments**

The authors have presented a very interesting idea for efficiently improving the estimates of 50-year extreme loads for wind turbines. Most attractive about the method is its overall simplicity, using the well known concept of importance sampling and deriving a simple gradient for the variance of the extrapolation estimate that can be used to add the required number of additional samples for each wind bin, in order to reduce the variance of the estimate to an acceptable level. With a few exceptions, to be noted below, the presentation of both background details and the method is clearly written. The

results seem to show that the method works quite well, without having to run the optimization for a prohibitively long time. However, there is one aspect of the setup of the problem that obscures the true meaning of the results, again to be expanded on below. Unless this can be clarified, interpretation of the results is difficult. As a preliminary demonstration of the approach, and assuming they hold up to scrutiny, these results are promising and the method could be of use both in conventional assessments of the 50-year extreme load and also in other settings like reliability assessments. It is also nice to see that the authors are open and critical about the overall power of the method and they make some interesting points about the limits of the current analysis framework and suggestions for other ways to improve extreme value analysis for wind turbines.

**Specific comments**

**Introduction:**

On page 2, lines 14-15 you write: "... reliable *extrapolation* of nonlinear physics under uncertain forcing is extremely problematic, especially without knowledge of the form (e.g. quadratic) of the nonlinearity." This is certainly true, but an an equally important reason why the specific type of long-term extrapolation usually done for the evaluation of 50-year extreme loads is problematic is precisely the large differences in timescale between the data and the extrapolated estimate. This impacts the problem in many ways, certainly also through the nonlinearity you mention, but in a more practical sense this large difference in timescales means that any uncertainty in the data is necessarily magnified by the extrapolation. Small errors in the short term data set could potentially lead a designer to significantly over- or underestimate the long-term extreme loads. Later in the paper you show that there can be a large variance in the extrapolation, which is in turn reduced by your proposed method, so this overall point should be mentioned here.

**Section 2.2, Extrapolation:**

On page 4, line 2 you write: "In this paper we use a 3 parameter Wiebull distribution." Why this distribution? This choice may for the demonstration of the method be seemingly irrelevant, but some reason for the choice would be instructive for the reader. One might wonder, for example, if this is indeed the distribution that overall gives the best fit for your data and is therefore the easiest to use for illustrating the method. Certainly, the Gumbel distribution, for instance, can be easier to work with (since it has only 2 parameters), so it seems there must be some motivation for choosing the Weibull. For reasons of clarity and reproducibility, it would also be of interest to know what method you used to fit the distributions to your data. Maximum likelihood estimation perhaps? Please state this.

On page 4, line 29 you write: "To gather peaks, we take the maximum of each 1 minute segment in our simulations." Yet, you already stated on lines 17-20: "Finally, there is the length of time between independent peaks ... values as low as 4 seconds have been justified in previous studies (Ragan and Manuel, 2008), and 10 seconds seems to be more than 20 adequate." So why this choice of 1 minute separations? Based on what you write later in the paper it seems to be motivated by a desire to only use the largest peaks, hence using smaller separations would yield maxima that might not give a good description of the extreme behavior of the system. If this is the case, or if there is some other motivation, it would be instructive to have it stated clearly here. To be very precise, given that one wants a number of peaks that exactly divides the total simulation length, why not a 30 second separation, which would give twice the number of maxima and hence more data per simulation with which to fit the distributions while maintaining 3 times the required separation to maintain independence of the peaks?

**Section 3.2, ASIS as stochastic optimization:**

As a more general point, it is clear from the fact that the variance is what is being minimized, as well as studying equation (21) and from the algorithm summary on page 10, that the procedure is only ever going to *add* samples to the various bins, never remove samples. However, viewed as a more general optimization problem, an unattentive

reader could believe that such an algorithm might in fact reduce the number of samples in a bin. Perhaps this "uni-directionality" of the algorithm should be stated more clearly to avoid any possible confusion over what its purpose is.

On page 10, lines 11-12 you write: "Also, the $N_i$ need to be integers, which is accomplished by rounding. The resulting error is likely subsumed into the general convergence of the stochastic optimization procedure." This statement needs a more convincing justification. For a k-dimensional optimization problem, which for a discrete solution set induces a k-dimensional lattice of discrete points, it is not immediately clear that a local minimum in the continuous case is also a local minimum in the discrete case. That is, when going from some continuous set of $N_i$ (a point in k-space) to a discrete set of $N_i$ (a point on the k-lattice) by rounding each $N_i$ to the nearest integer, the corresponding function value is not necessarily lower than at neighboring points on the lattice. If such a correspondence between minima in continuous and discrete space can be established for this particular case, it needs to be justified by specific arguments or at least by reference to another work.

On page 10, line 18 you write: "In fact we have to decide some number of "large peaks" we will use to evolve $N$." Why? Presumably you already have all the information from the extrapolation that has already been performed before estimating the gradient. So why must the gradient only be estimated from some limited number of large peaks? The motivation for this is certainly not clear from the text.

**Section 4, Results:**

To demonstrate the method, you set up a problem where you seek the ideal number of peaks, $M_{pks}$, to use for the fitting. However, it is unclear how this variable relates to the extrapolation procedure already described. Initially, you have used 10 maxima, 1 per minute of simulation time, for each wind speed bin. Is it this number of 10 maxima which is now a variable? If so, how is this actually accommodated in the extrapolation? For example, in Figure 2 you show the extrapolation for many different values of $M_{pks}$.

How do you get the additional peaks? Do you perform another T minutes of simulation corresponding to the number of additional peaks needed? Or do you decrease the separation time between maxima and hence extract more maxima from the simulation data you already have? If the former, do you perform specifically the required amount of simulation time for each case or do you perform enough simulations for all the different values of $M_{pks}$ you have used and then simply use the 5, 10, 20, 40 etc largest peaks from this expanded set of potential maxima? These different ways of solving the problem have very different statistics and therefore different implications for the extrapolation itself. Using the largest 20 maxima from 160 minutes of simulation time is very different from using 20 maxima from 20 minutes, which is very different from using the 20 largest maxima from 10 minutes and so on. In fact, depending on which of these approaches is used, it is not clear whether the results are truly meaningful. Nor is it clear how the number of maxima used might or might not interact with the number of samples in each bin as dictated by the ASIS optimization procedure.

More details about the optimization are needed in order to ensure that the results are clear and reproducible. For example, the criteria for termination of the algorithm are unclear.

**Technical corrections**

- Section 2.2, page 4, line 2: "In this paper we use a 3 parameter Wiebull distribution." Should be "Weibull"

- Section 3.1, page 8, equation (12): $Y(x)$ should be $Y(x_i)$

- Section 3.1, page 8, lines 14-15: "Here we denote the dataset as $\{Y_{i,k}\}$ where $i$ indexes over wind speed bins and $k$ indexes over the peaks we have extracted at that wind speed." This repeats almost exactly information already given in Section 2.2, page 3, lines 30-31. Consider removing (since we are already aware of the notation) or rephrasing.

- Section 4, page 11, Figure 2, caption: "... (a lower values... " should be "value"

---

## Author Response (AR1)

Reply to comments of "Referee #1", Lance Manuel:

First, the authors wish to thank Professor Manuel for his detailed and thoughtful comments. The reviewer's comments point out some lack of precision in our discussion, especially that of IFORM, and will serve to improve the paper considerably. Point by point replies follow:

```
The entries in "Courier" font have been added (Dec, 2017)
to our initial replies (Oct, 2017, "Times" font) as we
prepared our revised manuscript.  They contain more
specific descriptions of what we actually changed in the
revised manuscript.
```

**Overview and General Comments**

*This is a most interesting presentation of a vexing problem that has proven to be a challenge to wind turbine loads analysts for many years. The ideas developed by the authors and the narrative discussing the desire to "bridge" more conventional extrapolation methods and variance reduction techniques that go beyond brute-force Monte Carlo simulations are welcome. Casting the problem as an optimization problem, albeit without the usual formalisms, so as to adaptively improve estimates of long-term loads is done most effectively. Throughout, there are interesting insights and discussions that make for an illuminating reading and exposure to the essential issues.*

**Specific Section-by-Section Comments**

**Introduction**

*The description about IFORM on Page 2, Lines 12–13 should be clarified. More correctly, only with the environmental contour (EC) method which is the most commonly employed version of IFORM, one uncouples the "environment" from the "response," and global extremes of interest associated with a target return period are approximated by using the maximum response from among all response levels derived only for candidate environmental variables consistent with that return period (in other words, response variability given environmental conditions is neglected). That said, in cases such as the one described in this article, where the environment is described using only one random variable (wind speed), the EC method has limited use. The EC method is better suited when a pair of random variables (say, wind speed and turbulence intensity, or wind speed and wave height for offshore turbines) are included. When only one random variable defines the environment, the "environmental contour" for a target return period is reduced to a single wind speed. This has limited use because, for*

*instance, in the present example, for both side-to-side and fore-aft tower bending moments, the method will suggest that only rare and high wind speeds around cut-out that are associated with a 50-year return period need to be considered in turbine aeroelastic simulations. This will clearly lead to inaccurate 50-year fore-aft tower moments.*

The applicability of EC does not depend on the *number* of random inputs as much as their joint probability. Even with a large number of random inputs, we can envision response variability being the governing influence, i.e. a situation where the extreme load occurs at rather common environmental conditions. This issue is just especially manifest in the 1-variable case, where indeed the 50-year EC is just a single point.

```
We have mostly rewritten the IFORM section based on your
comments.  Thank you very much for helping clarify this
issue.
```

*Now, despite the preceding comment that the EC method reduces the conventional environmental contour to a degenerate point or single-valued wind speed to consider for turbine response simulations, in fact, in the present case, one could instead use IFORM in its more general form and use wind speed and response as two random variables and formally derive estimates of 50-year side-to-side and fore-aft tower bending moments. This is discussed later, along with comments offered in the context of Section 2.4 (IFORM).*

This is true, however, such estimates reintroduce extrapolation, which ASIS is designed to avoid (see below).

**Extrapolation**

*Page 4, Line 14: Strictly speaking, the 50-year return period event is that event that is exceeded "on average" once in 50 years. Even though it is not the same in general, sometimes the event is defined as one that is exceeded "on average" with a probability of 1/50 in one year.*

Thank you for this clarification; we should base our definition on the underlying assumption that this is a Poisson process and define our terms precisely from it.

```
Inserted comment to this effect.
```

*Page 4, Line 26: $3.8^{-7}$ should be $3.8 \times 10^{-7}$.*

Thank you, will be corrected.

```
Corrected
```

**Monte Carlo importance sampling for extreme loads**

*Page 6, Line 23: The comment that some form of accept-reject sampling can be used with importance sampling is an intriguing one. It is unclear how exactly this would be done given that Y (x) is not known in closed form; any additional notes, even if included very briefly, regarding such sampling would help.*

The comment is meant to address the problem that the normalization constant is not known, not that Y(x) is an expensive function. As long as we can *evaluate* Y(x), even if by simulation (we assume we can also evaluate f(x)), we can sample from any distribution proportional to Y(x)f(x) by the accept-reject algorithm (see, e.g. https://en.wikipedia.org/wiki/Rejection_sampling), which involves sampling uniformly in a 2D region containing the function Y(x)*f(x). The probability of x w.r.t. the Y*f distribution is just the proportion of these uniform samples below Y*f in this 2D "box". This procedure does require assumptions on the bounds of Y and the support of Y and f, but in principle these can be made large enough to "cover" any meaningful probability for Y*f.

```
We have inserted this explanation into the text for the
curious reader.
```

**IFORM**

*As stated earlier, the EC method doesn't apply here as there is no environmental contour corresponding to the authors' example—such a contour is a degenerate single wind speed value obtained as $F^{-1}(3.8 \times 10^{-7})$ and as such has limited value for, say, the fore-aft tower bending moment where the derived 50-year load will certainly be under-predicted. Indeed, in this single environmental random variable case, the degenerate single simulation needed for any response or load of interest would require simulations to be run for a single wind speed above cut-out, i.e., for V equal to 43.3 m/s. This would be meaningless.*

*The authors correctly point to the deficiencies of IFORM (on Page 7) but, in light of comments in the preceding paragraph, since there is no environmental contour at all that can be defined to describe their example study, much of the extended discussion regarding the EC method and environmental contours as presented in Section 2.4 is not relevant.*

The discussion of EC is in part simply for reasons of completeness, but more importantly it is for conceptual aid. For us to reach low probability events *without extrapolation* we need to correlate the extreme events directly to environmental conditions. Otherwise, we must *model* the response variability, which puts us back in the modeling and extrapolation context we are seeking to overcome.

We have substantially re-written the section on IFORM. We hope it is no longer misleading and irrelevant. We think the distinction between environment and response, whether in the context of IFORM, EC, or not, is an important one: quantifiable uncertainty in "environment" we can handle, while unquantified uncertainty in "response" we cannot. The latter must therefore be treated either by assuming the data comes from a certain distribution, fitting the data to that distribution, and extrapolating to desired return periods, or by massive numbers of Monte Carlo samples such that empirical CDFs reach the desired return period. IFORM (in its general sense) is simple and even beautiful to use, and it may be formulated differently than the traditional extrapolation approach, but as used it is still a form of extrapolation. It does not save us from the "extrapolate or sample massively" conundrum.

Another way to say this: as a *method*, IFORM is great, but it is a form of extrapolation, which we are trying to avoid by using Monte Carlo approaches; but as a *concept*, IFORM/EC adds tremendous value to the discussion, because it highlights the notion of response variability, which is the crux of why extreme loads estimation is so hard to pin down.

The conclusion of the paper, after all the method development, is that the best step forward is to study the actual response variability. If we can quantify this variation, then we can target specific quantiles. For example, suppose we *knew* that the side-side load was, say, a Weibull distribution with a certain shape and scale (these could of course be functions of wind speed). Then the EC method could be directly used to find all the combinations of wind speed and load that have the 50-year return period probability. Take that max load of these and we are done. All the fitting and extrapolation are necessary because we do not yet have an understanding of the response variability. Maybe we *cannot* have such an understanding, but trying seems justified, since that would be the only way to avoid extrapolation and to avoid massive computation, both of which everyone seems to agree is problematic.

The discussion of IFORM/EC in the paper is not meant to
really do justice to IFORM/EC as methods, but only to frame
this discussion of the critical distinction, statistically,
between environment and response.  Perhaps discussing it in
this way is unfair; there is no intent to dismiss IFORM/EC
as practical methods.

*Now, in a most interesting way, the very issue that doesn't allow for a critique of the EC method—namely, that the authors choose only V as an environmental random variable—actually allows the more general IFORM procedure to be used with the authors' own simulation results and will lead to reasonable results (how this can be achieved is presented here very briefly). The idea is as follows: Consider that there are two random variables—wind speed, V, and the response or load, Y , whose statistics are derived from 10-min simulations . We will assume that we know the probability distribution for V (for instance, here, the authors use a Weibull V with shape and scale parameters equal to 2 and 11.28 m/s, respectively); we establish conditional distributions for Y given V based on simulations. We can use IFORM, though not the EC method, to find the required quantiles of Y |V for any V of interest. This is purely a geometry problem (involving mapping of V and Y to two independent standard normal random variables). To illustrate this, because the results presented in Figure 1 are the easiest to read off and learn from without great effort, one would find using IFORM that for TwrBsMxt, the 24 m/s bin would require that the desired 50-year response must have a probability of exceedance in 10 minutes of $5.95 \times 10^{-6}$. Given the 5th, 25th, 50th, 75th, and 95th percentile loads in Figure 1, a 2-parameter Weibull fit to these data leads to a 50-year TwrBsMxt value of 29,700 kN-m. Other (lower) wind speeds occur more often and associated load levels to be checked for those wind speeds using the IFORM procedure must be rarer, i.e., with exceedance probabilities in 10 minutes that are smaller than $5.95 \times 10^{-6}$. Given the data, these wind speeds do not lead to TwrBsMxt values at the desired probability levels that exceed what was found for V equal to 24 m/s. In a similar manner, for TwrBsMyt, selecting V equal to 16 m/s, the desired 50-year response for IFORM must have a probability of exceedance in 10 minutes of $7.20 \times 10^{-7}$. Again, from the data in Figure 1, a 2-parameter Weibull fit to these data read off easily, leads to a 50-year TwrBsMyt value of 94,300 kN-m. Again, note that other wind speeds and associated (different) response quantiles need to be checked, as part of the IFORM procedure, to ensure that the largest load quantile across all the wind speed bins is then claimed as the 50-year load. Details regarding all the calculations are not presented here but IFORM computations are based on the Weibull V and loads data from Figure 1.*

This is wonderful!  You are the most inspired (and inspiring reviewer) ever.  We have no objection with your procedure, and it is certainly interesting that your results largely agree with ours.  We would point out, however, that in carrying out the IFORM procedure described above, you have taken our data and *fit* it to a Weibull distribution. The resulting low probability estimations are then made possible by *extrapolation* of this

fitted model. It would be interesting to investigate how this differs from simply fitting an extreme value distribution to the empirical data directly (as in the traditional bin-based IEC-recommended method). It is quite possible that the IFORM, even though still based on fitting and extrapolation, is fundamentally more accurate because in some sense it "factors out" the environmental probabilities. If some form of extrapolation is inevitable to get to 50-year loads in a tractable amount of computing time, maybe the combination of ASIS's variance-minimization-sampling and general IFORM (as you describe) is a promising approach.

*In sum, the authors' comment regarding searching on the environmental contour (or just inside it) is not pertinent here. There is also no need to discuss above-median response levels in this context. Both the preceding comments would have been appropriate if, in addition to wind speed, another environmental random variable were included such as turbulence intensity. As illustrated, IFORM in its general form can be easily employed here and response variability can be directly accounted for—as the authors state correctly, this variability is ignored by the EC method. It is not ignored by IFORM in general and, as shown in the previous paragraph, even with the limited data presented in Figure 1, the method works quite efficiently in deriving 50-year loads. By running additional simulations at critical wind speeds, the resulting loads data and subsequent distribution fits to the same will lead to reduced uncertainty in derived 50-year loads.*

Again, this is perhaps an important "intermediate step" between the traditional bin-based extrapolation method and the fully model-free ASIS method, i.e. more accurate than direct extrapolation but more computationally tractable than ASIS. But we feel it is important to point out that it does still indeed rely on fitting and extrapolation to achieve the desired 50-year return periods. Filling this gap precisely would be a very interesting area for future study.

```
By optimizing the sampling, ASIS can benefit any method
that fits data to distributions, so certainly an ASIS-IFORM
hybrid would be of interest.
```

**Adaptive stratified importance sampling (ASIS)**

*Reference to POE, on Page 8, Line 20, should really be to the probability of non-exceedance.*

Indeed correct; we can easily correct this.

```
Done
```

*In Step 2 of the algorithm on Page 10, why not simply obtain new samples (with rounding) in proportion to $\Delta_N J(N)$? It is not clear but it seems that the most important bin (where $\Delta_N J(N)$ is largest) is allocated some samples and the remaining (of 20) are then*

*randomly allocated to other bins. Why randomly?*

Randomly because we cannot strictly rely on information gained from the limited number of samples gathered so far. Due to random variation, the first N samples might *not* be leading us toward the correct bins, so "following" them could lead to a *local* solution to the minimal variance problem. Choosing the rest of the samples randomly is a crude (but common) strategy in global optimization.

```
Language that attempts to clarify this situation has been
added below the algorithm description.
```

*The algorithm, as presented, appears not to state what is the criterion for stopping or convergence. The discussion regarding the "umbrella" concept that suggests a minimal superset of sample distributions across bins is exactly what is needed. It is the only way to guarantee adequate samples of response extremes to meet very distinct response characteristics such as between TwrBsMxt and TxrBsMyt. The ASIS algorithm as presented doesn't explicitly state this but assumes convergence when all the response measures are adequately sampled in all bins so as to yield unbiased 50-year response values, presumably with some specified confidence level on these predictions. If it helps, in an offshore wind turbine application, Sultania and Manuel [3], employed bootstrap-based confidence intervals for specific sea states (akin to bins here) to arrive at the appropriate number of simulations for accuracy in response probability distributions, conditional on the environment. Given the results presented in this article, it appears that convergence on long-term loads for each bin is indeed achieved by the authors in their examples by examining the variance of loads associated with low POE levels.*

The lack of precise convergence criteria is indeed a shortcoming of the present paper, especially because in practice this is critical: the user is seeking an estimate of the 50-year load with some acceptable measure of its accuracy. We have pointed at the way one would achieve this in practice. In particular, we would recommend (as in the "Results" section) to use ASIS iteratively in conjunction with extrapolation and bootstrapping: For each ASIS iteration, subsample via bootstrapping, form a large number of extrapolations, thus estimate the 50-year load *and its variance*. The stopping criteria is then a user-specified threshold for the variance.

```
The lack of true stopping criteria in the current work is
because we are still investigating the major properties of
the method. We have added a recommendation and citations
and recipe to use bootstrapping for production purposes.
```

*It was not clear, upon reading, what was the reason for using the 5 largest loads. The largest loads will automatically drive the tail of long-term loads distributions and, as such, the 50-year load, when these largest loads are included along with all smaller loads; so, why retain only the 5 largest? Some clarification would help here.*

This is indeed another part of the algorithm that would need further study before committing it to "production" use. Calculating the gradient of the variance using the 5 largest peaks was just an intuitive guess as to the number of loads that would drive the sampling in an effective way. A similar parameter is the number of peaks used for extrapolation $M_{pks}$, which we studied (at least graphically) in Figures 2, 4, and 5. Future work would be warranted to tune these parameters more systematically; hopefully it is convincing that the exact values of them does not undermine the principle of the method.

```
Language intended to clarify this situation has been added.
Using all the peaks would reduce the variance of the POE
estimates of all the peaks, even the small peaks we don't care
about, but using just the single largest might miss the (as yet
undiscovered) peaks associated with other bins. The choice of
the "5 largest peaks" is rather arbitrary, enough so that more
than the single largest peak contributes, but not so many as to
deemphasize the goal of finding large peaks.
```

**Results**

*The results in Figures 3–5 are very interesting and suggest that the methodology proposed by the authors offers a robust and efficient means of deriving long-term turbine loads for design. The rapid reduction in variance in estimates of long-term loads by using the ASIS algorithm is convincing.*

*A few observations are offered. 1. It would be very useful and insightful to see how g(x) or the bin-wise importance sampling changes with iteration for the two load measures, TwrBsMxt and TwrBsMyt, separately, and what the ultimate umbrella sampling ends up being, after convergence, or as the number of iterations changes—given the contrasting characteristics of the two load measures, one might expect a bi-modal sampling with one mode around or slightly above rated and another closer to cut-out will result.*

In our earlier paper ("Advances in the Assessment of Wind Turbine Operating Extreme Loads via More Efficient Calculation Approaches" in: AIAA SciTech 2017) we have (see Figure 6 therein) plotted the distributions that ASIS led to. In that paper, the ASIS method was applied separately to each load, and they do result in bin distribution peaks at different wind speed values. So it would indeed be predicted that a bimodal distribution would emerge.

```
In fact, when we dig into the data from our runs, we find
that a bi-modal distribution does not emerge, despite our
intuition. It would indeed be interesting to do more
detailed comparisons of the distributions that emerge as a
function of loads that are included. Sometimes algorithms
do not conform to our intuition!
```

*2. It is not completely clear what $M_{pks}$ refers to. For instance, does $M_{pks} = 40$ imply that, after accounting for samples from all bins, only the largest 40 are used or does it mean that for each bin, the largest 40 loads are retained and then combined with 40 from the other bins before extrapolation?*

The $M_{pks}$ largest loads from each bin were used to separately fit the extreme value distributions in a bin-wise fashion.

In the results section, we have expanded the description of exactly what was done, especially what exactly $M_{pks}$ is.

*3. The non-monotonic upward trending cooefficient of variation on the 0.05 POE load seen in Figure 3 for a very large number of peaks might be caused by dependence among the peaks that results when too many peaks are extracted from each 10-min simulation (see Fogle et al [2]).*

Indeed, this is likely, especially because the effect is most pronounced for the lowest iterations where there are fewer total runs to choose from.

*4. Is it possible that the x axis units in Figure 4 are 10,000's of KN-m. rather than 1,000's of kN-m as stated? This would be consistent with Figures 1 and 2 which appear to show TwrBsMxt loads an order of magnitude higher. It would also be consistent with the IFORM-based estimates that were computed (above) as $29.7 \times 10^3$ kN-m and $94.3 \times 10^3$ kN-m, respectively, for TwrBsMxt and TwrBsMyt. The caption for Figure 5 is correct; the one for Figure 4 might need to be corrected.*

Agreed. Again, thank you for reading carefully!

Corrected.

*5. As presented, Figures 4 and 5 do not show loads for the low POE level of $3.8 \times 10^{-7}$ associated with the target 50-year return period. It would have been useful to see those results. Indeed, ASIS-based convergence at the lowest POE levels that are presented, suggest that 25 iterations and the use of 40 peaks is very good.*

It would of course be possible to extrapolate to the 50-year levels in practice. Though ASIS appears to work well, we see there is no free lunch; it would still require many iterations to achieve empirical 50-year estimates. Apologies for not extending the extrapolation all the way to the 50 year return period; in our opinion the figure was visually more appealing as presented, because it illustrates more clearly the improvement made over the 25 iterations of ASIS.

*In the caption for Figure 3, $5^{-2}$ should be $5 \times 10^{-2}$.*

Thank you for point that out; we will correct this.

`Corrected`

**Conclusions**

*The closing discussion is most helpful in setting this work in the context of other studies regarding the derivation of long-term loads for wind turbine design. The ASIS algorithm, as presented, will prove useful to loads analysts. While the authors have demonstrated its effectiveness using 100 independent tests in their numerical studies, in practice, it may be useful to suggest use of the ASIS algorithm followed by bootstrapping, after each iteration, and confirmation that the coefficient of variation on some load quantile (as in Figure 3, for example) is acceptably low—for example, 5%. It is clear that ASIS can achieve this target with a far smaller number of aeroelastic simulations than extrapolation based on ordinary or conventional sampling methods. The authors do, in fact, recognize in their closing discussions, that bootstrapping with a single data set could be used with ASIS and thus reduce computational effort.*

*Only to provide a contrast with the authors' work, it should be noted that IFORM-based approaches, referred to in this article, can be very efficient. This has been demonstrated in this discussion using the authors' own data (crudely derived from their Figure 1); note that in the illustration presented response variability is not ignored as is done with the environmental contour method. Accounting for response variability in IFORM is not difficult—it is conceivable that an ASIS-like formulation for sampling could prove far more efficient than even IFORM but this needs to be demonstrated for situations where more than one random variable defines the "environment."*

It is possible, as noted above, that though IFORM in this context still involves extrapolation, that it is fundamentally more accurate than direct fitting to extreme value distributions, because the environmental contour has already been accounted for. Thus, an improvement on the recommend ASIS+extrapolation+bootstrap approach might instead be ASIS+IFORM+bootstrap. This would be a subject of future study.

*Finally, related to this last point, the ASIS algorithm would be especially important to employ along with the introduction of stochastic turbulence and with the treatment of turbulence intensity or turbulence standard deviation explicitly as a random variable. The role of gusts and turbulence in extreme loads is known to be significant and the need then for bivariate importance sampling distributions could present challenges (Bos et al. [1], van Eijk et al. [4]); at the same time, the benefits to be derived from approaches such as ASIS, if its efficiency and accuracy is demonstrated in such cases, would then add to its appeal.*

This would be another interesting area of future study/application. In principle it is a simple extension of the method to apply the stochastic variance minimization procedure

to more than one variable.

A suggestion to this effect has been added to the conclusion.

*On Page 15, Line 3, "stratified adaptive" should be "adaptive stratified" to be consistent with the ASIS acronym.*

Yes.

Corrected.

Finally, thank you very much once again for the incredibly careful and thoughtful reading of the paper.

Yes, thank you again. A most thorough review!

On behalf of the co-authors,

Sincerely,

Peter Graf,
Computational Science Center and National Wind Technology Center, National Renewable Energy Laboratory

Reply to comments of "Anonymous Referee #2":

First, the authors wish to thank "Referee #2" for the detailed and thoughtful comments. The reviewer's comments will serve to improve the paper considerably. Point by point replies follow:

```
The entries in "Courier" font have been added (Dec, 2017)
to our initial replies (Oct, 2017, "Times" font) as we
prepared our revised manuscript.  They contain more
specific descriptions of what we actually changed in the
revised manuscript.
```

**General comments**

*The authors have presented a very interesting idea for efficiently improving the esti- mates of 50-year extreme loads for wind turbines. Most attractive about the method is its overall simplicity, using the well-known concept of importance sampling and deriving a simple gradient for the variance of the extrapolation estimate that can be used to add the required number of additional samples for each wind bin, in order to reduce the variance of the estimate to an acceptable level. With a few exceptions, to be noted below, the presentation of both background details and the method is clearly written. The results seem to show that the method works quite well, without having to run the optimization for a prohibitively long time. However, there is one aspect of the setup of the problem that obscures the true meaning of the results, again to be expanded on below. Unless this can be clarified, interpretation of the results is difficult. As a preliminary demonstration of the approach, and assuming they hold up to scrutiny, these results are promising and the method could be of use both in conventional assessments of the 50-year extreme load and also in other settings like reliability assessments. It is also nice to see that the authors are open and critical about the overall power of the method and they make some interesting points about the limits of the current analysis framework and suggestions for other ways to improve extreme value analysis for wind turbines.*

**Specific comments**

**Introduction:**

*On page 2, lines 14-15 you write: "... reliable extrapolation of nonlinear physics under uncertain forcing is extremely problematic, especially without knowledge of the form (e.g. quadratic) of the nonlinearity." This is certainly true, but an equally important reason why the specific type of long-term extrapolation usually done for the evaluation of 50-year extreme loads is problematic is precisely the large differences in timescale between the data and the extrapolated estimate. This impacts the problem in many ways, certainly also through the nonlinearity you mention, but in a more practical sense this large difference in timescales means that any uncertainty in the data is necessarily magnified by the extrapolation. Small errors in the short-term data set could potentially lead a designer to significantly over- or underestimate the long-term extreme loads. Later in the paper you show that there can be a large variance in the extrapolation, which is in turn reduced by your proposed method, so this overall point should be*

*mentioned here.*

This is an excellent point about the difficulties due to time scales and we can certainly add some language to that effect to the problem introduction.

`We have added a paragraph expressing the difficulty in terms of`
`timescales as suggested.`

***Section 2.2, Extrapolation:***

*On page 4, line 2 you write: "In this paper we use a 3 parameter Wiebull distribution." Why this distribution? This choice may for the demonstration of the method be seemingly irrelevant, but some reason for the choice would be instructive for the reader. One might wonder, for example, if this is indeed the distribution that overall gives the best fit for your data and is therefore the easiest to use for illustrating the method. Certainly, the Gumbel distribution, for instance, can be easier to work with (since it has only 2 parameters), so it seems there must be some motivation for choosing the Weibull. For reasons of clarity and reproducibility, it would also be of interest to know what method you used to fit the distributions to your data. Maximum likelihood estimation perhaps? Please state this.*

We agree there is a lack of a full explanation here. We used the 3-parameter Weibull because it has worked well in previous work. There are many excellent papers concerning extrapolation and efforts to justify and/or distinguish between different extreme value distributions (e.g., Ragan and Manuel,"Statistical Extrapolation Methods for Estimating Wind Turbine Extreme Loads", Toft, Sorenson, and Veldkamp, "Assessment of Load Extrapolation Methods for Wind Turbines"). It is not the purpose of the current paper to assess the different distributions.

`We have added language specifically stating that we are *not*`
`claiming any superiority for the 3 parameter Weibull.`

Regarding the fitting procedure, in light of the interest in particular in extrapolation, rather than just fitting, we have fit the empirical cumulative distribution function of the data directly to the theoretical cumulative distribution function (CDF) of the distribution by nonlinear least squares. We have done this separately for the data from each wind speed bin. In order to emphasize the largest peaks (i.e. the lowest probability values) we do not use all the data, just the $M_{pks}$ largest peaks, where $M_{pks}$ is an algorithmic parameter. As an exercise, we experimented with using different values of $M_{pks}$ (e.g., see Fig 2).

`We have added language describing this procedure.`

*On page 4, line 29 you write: "To gather peaks, we take the maximum of each 1 minute segment in our simulations." Yet, you already stated on lines 17-20: "Finally, there is the length of time between independent peaks ... values as low as 4 seconds have been justified in previous studies (Ragan and Manuel, 2008), and 10 seconds seems to be more than 20 adequate." So why this choice of 1 minute separations? Based on what you write later in the paper it seems to be motivated by a desire to only use the largest peaks, hence using smaller separations would yield maxima that might not give a good description of the extreme behavior of the system. If this is*

*the case, or if there is some other motivation, it would be instructive to have it stated clearly here. To be very precise, given that one wants a number of peaks that exactly divides the total simulation length, why not a 30 second separation, which would give twice the number of maxima and hence more data per simulation with which to fit the distributions while maintaining 3 times the required separation to maintain independence of the peaks?*

The tradeoff between gathering more peaks (at the risk of sacrificing statistical independence) versus less peaks (at the risk of not having enough data) is another interesting detail that we could study. In this vein, our use of 1 minute intervals and your proposal to use 30 second separations has a similar motivation, which is simply to pick a reasonable point along that tradeoff that avoids the pitfalls of either end point.

```
Language to this effect has been added to the manuscript.
```

**Section 3.2, ASIS as stochastic optimization:**

*As a more general point, it is clear from the fact that the variance is what is being minimized, as well as studying equation (21) and from the algorithm summary on page 10, that the procedure is only ever going to add samples to the various bins, never remove samples. However, viewed as a more general optimization problem, an unattentive reader could believe that such an algorithm might in fact reduce the number of samples in a bin. Perhaps this "uni-directionality" of the algorithm should be stated more clearly to avoid any possible confusion over what its purpose is.*

Agreed. In principle, we can imagine "solving" the bin-optimization problem once and for all, which would give us an optimal distribution of bins. Then, given a certain computational budget, we could apportion the samples to bins proportionally, which would indeed possibly reduce the number in some bin from the original search. Our orientation was more from an "online" perspective: why not use data from bins you have already run simulations on? It would be easy to add a sentence clarifying this point.

```
Added language to this effect.
```

*On page 10, lines 11-12 you write: "Also, the $N_i$ need to be integers, which is accomplished by rounding. The resulting error is likely subsumed into the general convergence of the stochastic optimization procedure." This statement needs a more convincing justification. For a k-dimensional optimization problem, which for a discrete solution set induces a k-dimensional lattice of discrete points, it is not immediately clear that a local minimum in the continuous case is also a local minimum in the discrete case. That is, when going from some continuous set of $N_i$ (a point in k-space) to a discrete set of $N_i$ (a point on the k-lattice) by rounding each $N_i$ to the nearest integer, the corresponding function value is not necessarily lower than at neighboring points on the lattice. If such a correspondence between minima in continuous and discrete space can be established for this particular case, it needs to be justified by specific arguments or at least by reference to another work.*

We have no such formal justification for this procedure. The thinking is simply that we assume the variance of the estimate as a function of bin distribution is a smooth enough function that it is a reasonable *approximation* to round to the nearest bin counts. Given that we do have to round

to integer bin counts, this is the best we can do. Furthermore, this is a stochastic problem, so it is likely that the continuous-valued bin counts have some error associated with them anyway. We claim (without proof, but believe it is a reasonable assertion) that the error induced by the unconverged sampling procedure is just as much or more than that induced by the rounding to integers of the bin counts. We are certainly not claiming that the rounded minimum is lower than the "relaxed" (i.e. continuous-valued) minimum; this is patently false, because the possible discrete bin-counts are a subset of the continuous ones.

```
Added language and reference connecting our algorithm with both
stochastic gradient descent and discrete programming. Hopefully
this is acceptably clear.
```

*On page 10, line 18 you write: "In fact we have to decide some number of "large peaks" we will use to evolve N." Why? Presumably you already have all the information from the extrapolation that has already been performed before estimating the gradient. So why must the gradient only be estimated from some limited number of large peaks? The motivation for this is certainly not clear from the text.*

This is, operationally, a separate step from the extrapolation (in fact, the ASIS variance minimization can be performed without ever performing extrapolation). In retrospect, there may be an interesting connection between the number of peaks used for extrapolation ($M_{pks}$) and the number of peaks that enter into the gradient-of-variance calculation, but this is beyond the scope of the current paper (though if you are confused, we will clarify this distinction in the text). The choice of the "5 largest peaks" is rather arbitrary, enough so that more than the single largest peak contributes, but not so many as to deemphasize the goal of finding *large* peaks.

```
We have attempted to clarify this issue in the text.
```

**Section 4, Results:**

*To demonstrate the method, you set up a problem where you seek the ideal number of peaks, $M_{pks}$, to use for the fitting. However, it is unclear how this variable relates to the extrapolation procedure already described. Initially, you have used 10 maxima, 1 per minute of simulation time, for each wind speed bin. Is it this number of 10 maxima which is now a variable? If so, how is this actually accommodated in the extrapolation? For example, in Figure 2 you show the extrapolation for many different values of $M_{pks}$. How do you get the additional peaks? Do you perform another T minutes of simulation corresponding to the number of additional peaks needed? Or do you decrease the separation time between maxima and hence extract more maxima from the simulation data you already have? If the former, do you perform specifically the required amount of simulation time for each case or do you perform enough simulations for all the different values of $M_{pks}$ you have used and then simply use the 5, 10, 20, 40 etc largest peaks from this expanded set of potential maxima? These different ways of solving the problem have very different statistics and therefore different implications for the extrapolation itself. Using the largest 20 maxima from 160 minutes of simulation time is very different from using 20 maxima from 20 minutes, which is very different from using the 20 largest maxima from 10 minutes and so on. In fact, depending on which of these approaches is used, it is not clear*

*whether the results are truly meaningful. Nor is it clear how the number of maxima used might or might not interact with the number of samples in each bin as dictated by the ASIS optimization procedure.*

Clearly the description of the choice of peaks to use for fitting is inadequate. For Figure 2, for example, we have run FAST for 10 minutes 20 separate times for each bin (the figure shows only the results for the 20 m/s bin, but the others are similar). This results in 200 minutes of total simulation time, thus according to our 1-peak-per-minute convention (which is fixed throughout the paper), 2000 peaks. Each line on the figure (e.g. "5 peaks", "10 peaks", etc.) is the result of fitting the 3 parameter Weibull CDF to just the largest $M_{pks}$ (e.g. 5, 10, etc) of these 2000. The line labelled "empiricial CDF" is the empirical CDF of the 2000 peaks. We were hoping to *avoid* having to argue about the distinctions you mention above by choosing very "vanilla", uncontroversial values for these types of parameters, but our lack of clarity has unfortunately brought them all back into play. We apologize and will clarify in the next draft. We do not think the choices we made effected the meaningfulness of the results. The number of maxima used is similar to the choice of threshold in peak-over-threshold method, or the simulation time in methods just taking the single maximum from each simulation. In this context it is well-studied in the literature, but we agree that its role in ASIS is a new twist. Figures 3, 4,and 5 suggest that it is not a completely critical issue, however. Quite a range of values for $M_{pks}$ are effective, and the effectiveness (e.g., see Fig 3) is a relatively (for stochastic optimization) smooth function of $M_{pks}$.

```
We have expanded the description of the procedure as outlined
above.  We hope it is understandable now.
```

*More details about the optimization are needed in order to ensure that the results are clear and reproducible. For example, the criteria for termination of the algorithm are unclear.*

The lack of precise convergence criteria is indeed a shortcoming of the present paper, especially because in practice this is critical: the user is seeking an estimate of the 50-year load with some acceptable measure of its accuracy. We have pointed at the way one would achieve this in practice. In particular, we would recommend (as in the "Results" section) to uses ASIS iteratively in conjunction with extrapolation and bootstrapping: For each ASIS iteration, subsample via bootstrapping, form a large number of extrapolations, thus estimate the 50-year load *and its variance*. The stopping criteria is then a user-specified threshold for the variance. In short, monitor the (left and middle for ASIS and ASIS+extrapolation, respectively) curves in Figure 2 until the variance is below the desired level.

```
The lack of true stopping criteria in the current work is
because we are still investigating the major properties of
the method. We have added a recommendation and citations
and recipe to use bootstrapping for production purposes.
```

***Technical corrections***

- *Section 2.2, page 4, line 2: "In this paper we use a 3 parameter Wiebull distribution." Should be "Weibull"*

- *Section 3.1, page 8, equation (12): $Y(x)$ should be $Y(x_i)$*

- *Section 3.1, page 8, lines 14-15: "Here we denote the dataset as $\{Y_{i,k}\}$ where i indexes over wind speed bins and k indexes over the peaks we have extracted at that wind speed." This repeats almost exactly information already given in Section 2.2, page 3, lines 30-31. Consider removing (since we are already aware of the notation) or rephrasing.*

Thank you for these.

`All corrected.`

And thank you again for your careful reading and consideration of the paper.

`Yes, thank you very much. Your review helped make the paper (we`
`hope) much more comprehensible.`

On behalf of the co-authors,

Sincerely,

Peter Graf,
Computational Science Center and National Wind Technology Center, National Renewable
Energy Laboratory

---

## Referee Report (RR1)

**Review: Adaptive stratified importance sampling: hybridization of extrapolation and importance sampling Monte Carlo methods for estimation of wind turbine extreme loads**

This paper proposes an approach for estimating wind turbine extreme loads by integrating the importance sampling method and extrapolation technique. In the past, extrapolation has been widely used for estimating extreme loads, but existing extrapolation methods do not consider the unequal importance of different wind speed bins on the estimation accuracy. Built on the importance sampling, the authors propose to use unequal sample sizes in different bins according to their importance in terms of variance minimization. The proposed approach is interesting in that it improves existing extrapolation methods through the adaptive sampling procedure. I have some comments to improve the methodology, notations and presentations.

1. First of all, notations are confusing. The quantity to be estimated (e.g., $P(Y < l)$) is different from its estimator. For the estimator, "hat" is conventionally used above the quantify (e.g., $\hat{P}(Y < l)$).

2. For clarity, I suggest that the estimator to estimate POE needs to be explicitly defined in a mathematical form. Here, the authors use the iterative method, so the estimator needs to incorporate the iteration index.

3. Even though the summary of the procedure is given in Section 3.2, this procedure does not contain the detailed information to implement the approach. For example, in Step 4, how to update the empirical estimates and extrapolation estimates? For extrapolation, will the data from the last iteration be used, or will data obtained from all iterations used?

4. The goal of this paper is to develop methods that make unbiased estimates while minimizing variance. But extrapolation cannot guarantee the unbiased estimation, as it approximates the conditional density with statistical models (3 parameter Weibull distribution in this paper). Even though the statistical models can be improved throughout iterations, the models are still surrogate models. Therefore, it should be discussed/justified how the proposed approach can provide a unbiased estimation.

5. It is not clear how $g(x_i)$ becomes $\frac{N_i}{N_{tot}}\frac{1}{\Delta x_i}$. Because $g(x_i)$ is a density, $\int g(x)dx$ should be 1, which is not in the proposed form. The authors might consider $g(x_i) \propto \frac{N_i}{N_{tot}}\frac{1}{\Delta x_i}$.

6. The most important question is what is the benefit of the proposed approach over the stratified sampling and importance sampling. Stratified sampling provides the closed-form of optimal $N_i$'s. So, it is not clear why one needs to use the stochastic optimization approach combined with the importance sampling in Sec. 3.2. Also, what is the benefit of the proposed approach over the method proposed in Choe et al. (2016) (or their prior study) that uses importance sampling only?

7. In the implementation results, the relative standard deviations are presented. To show whether the estimates are unbiased or not, the POE estimates (or extreme load estimates) from the approach should be compared with those from crude Monte Carlo (e.g., authors may compare the estimates with those in the following paper: Barone, M., Paquette, J., Resor, B., and Manuel, L., 2012, "Decades of Wind Turbine Load Simulation," AIAA Paper No. 2012–1288.)

---

## Referee Report (RR2)

*Discussion of:*

**Adaptive stratified importance sampling: hybridization of extrapolation and importance sampling Monte Carlo methods for estimation of wind turbine extreme loads**

Nikolay Dimitrov
nkdi@dtu.dk
04 March 2018

The authors present a novel, hybrid approach to statistical load extrapolation. The method employs the importance sampling theory for estimating the contribution of samples in different wind speed bins to the variance of the extrapolated values. The result is an adaptive, stratified importance sampling procedure which converges faster than the traditional method of stratified sampling.

The article is well written and easy to read. The formulas seem free of technical errors. Some parts of the method may need a more rigorous description though – see the specific comments.

I think the article will also benefit from expanding some of the discussions and referring to additional literature. The authors may want to consider the publications listed as references in this document. Some further details about the references follow in the comments below.

**General comments**

1) As the authors say on Page 2, line 9: "Monte Carlo methods: exceedance probabilities are written as expectations of indicator functions". This is in fact a passing definition for structural reliability problems, regardless of the method used for the integration (the computation of the expected value). We can therefore make a parallel between the statistical extrapolation and the structural reliability analysis problems. This means we can take inspiration from literature where e.g. adaptive importance sampling is used as a tool for reliability analysis. Some examples include [1], [2] where search-based (adaptive) importance sampling is formulated and also [3] where it is applied to a dynamical system.

2) The current results section mainly shows the behaviour of the ASIS algorithm, but it is impossible to judge how it will compare to standard extrapolation procedures using the same number of FAST simulations. The algorithm suggested by the authors is essentially an approach for "guided" sampling from the joint distribution of environmental conditions. It would be really interesting to see a comparison with e.g. direct sampling from the long-term distribution using a pseudo Monte-Carlo simulation with low-discrepancy series, followed by extrapolation directly from the empirical CDF of the long-term load distribution.

3) Statistical extrapolation is a very academic exercise. In the last years the wind energy scientific society has published multiple papers which present attempts at improving the efficiency and reducing the uncertainty in load extrapolation. But the majority of these solutions represent highly complex algorithms with many parameters to tune and with tricky implementation which can easily

be done in a wrong way. My general concern with statistical extrapolation is that this complexity hinders the adoption of new research in industry as it requires a significant level of very specific expertise in order to achieve a correct and robust implementation. As a confirmation to this observation, the coming IEC 61400-1 ed.4 standard actually suggests methods for doing extrapolation with even lower complexity (and most probably higher uncertainty) than what was suggested in ed.3 of the same standard. This should not be viewed as a criticism to the present paper or its authors – but in my opinion the focus of future research in statistical extrapolation should not only be towards improving accuracy or efficiency, but also towards achieving robust and easy to implement solutions. This is something the authors may want to give a thought to and eventually discuss in their paper.

**Specific comments**:

4) Page 5, line 24: The authors use 10 peaks per 10-minute simulation. However, they do not seem to take any measures for ensuring statistical independence between the peaks (or there is no description of any measures they have taken). Thus, there is a very high chance that some of the events they have identified as peaks are correlated (e.g. they have small time separation). The result is most likely a (slight) over-prediction of the exceedance probabilities. A simple and straightforward measure for reducing dependency could be enforcing some minimum time separation between successive 1-minute peaks. Typically time separation equivalent to the time for 1-2 rotor revolutions should be sufficient to eliminate dependence almost entirely.

5) Figure 1: The amount of variability in the peaks is also a consequence of the authors' choice to use one-minute peaks. Drawing the same plot for the 120 ten-minute peaks per bin would show lower variation.

6) Section 2.2: Some good additional references to consider in this section are [4] which presents an excellent method for accounting for independence between peaks, [5] which is an extensive parametric study on extrapolation considering, among other things, the effect of number of peaks on extrapolation accuracy, and [6] where the seed-to-seed variability is addressed explicitly, though with focus on fatigue loads.

7) Section 2.4: I think this section can be removed or shortened significantly without reducing the quality of the paper. To me, what the authors try to define in the section is that the extrapolation per wind speed represents a set of conditional approximations which can best be described by a Rosenblatt transformation [7]. The IFORM method is related to the Rosenblatt transformation as the Rosenblatt transformation is a convenient way to draw the IFORM contours. Further, the IFORM method is meant to be applied to static quantities as e.g. 10-minute statistics of environmental conditions. The inventor of the method (S. Winterstein) has also worked on re-formulating the IFORM for dynamic problems as the one considered by the authors of the present paper [8]. Another relevant method is the tail-equivalent linearization method (TELM) [9], which employs FORM for approximation of the extreme response of dynamical systems. In general my feeling is that discussing FORM or IFORM in the current version of this paper is a distraction from the main scope – but if it is necessary to retain it, the TELM method could be a possible bridge.

8) Page 11, lines 15-25: As the authors state, this algorithm will converge to a local optimum, and measures should be taken to ensure that the global optimum is found. We could again make the parallel to structural reliability, where multiple failure modes lead to a system reliability problem

with multiple design points ("local minima") and the failure domain may be non-convex. When employing (adaptive) importance sampling, the system reliability problem is normally approached by using multiple "seeds" of importance sampling densities which are initially placed in the different parts of the variable domain. Melchers [2] for example suggests using stratified sampling or purely random sampling to generate the set of initial locations of the importance sampling densities. The authors may want to consider such an approach for their problem – it can potentially also help with the issue of the largest variance being at different wind speeds for different load channels.

9) Figure 2: A good reference for the effect of number of peaks (time series) on the extrapolation is [5], where this is investigated for several types of extrapolation methods, distribution fits, and with up to 1000 minutes of time series per extrapolation. Another relevant study is the one by Zwick & Muskulus [6] where they thoroughly investigate the seed-to-seed variability problem, though without considering extrapolation.

**Technical comments**:

10) Page 7, line 28: Y*f] is probably a typo. Did the authors mean [Y*f]?

**References**:

[1] Au, S. K. & Beck, J. A new adaptive importance sampling scheme for reliability calculations. *J. Struct. Saf.* **21,** 135–158 (1999).

[2] Melchers, R. E. Search-based importance sampling. *J. Struct. Saf.* **9,** 117–128 (1990).

[3] Mori, Y. & Ellingwood, B. Time-dependent system reliability analysis by adaptive importance sampling. *Struct. Saf.* **85,** 1–9 (1993).

[4] Naess, A. & Gaidai, O. Estimation of extreme values from sampled time series. *Struct. Saf.* **31,** 325–334 (2009).

[5] Dimitrov, N. Comparative analysis of methods for modelling the short-term probability distribution of extreme wind turbine loads. *Wind Energy* **19,** 717–737 (2016).

[6] Zwick, D. & Muskulus, M. The simulation error caused by input loading variability in offshore wind turbine structural analysis. *Wind Energy* **18,** 1421–1432 (2015).

[7] Rosenblatt, M. Remarks on a Multivariate Transformation. *Ann. Math. Statist*. **23** (3), 470-472 (1952).

[8] Lutes, L. D. & Winterstein, S. R. Design Contours for Load Combinations: Generalizing Inverse FORM Methods to Dynamic Problems. *Proceedings, 7th Comput. Stoch. Mech. Conf.* (2014).

[9] Fujimura, K. & Der Kiureghian, A. Tail-equivalent linearization method for nonlinear random vibration. *Probabilistic Eng. Mech.* **22,** 63–76 (2007).

---

## Author Response (AR2)

Dear Reviewer ("referee 2"),

Thank you very much for reviewing our paper, "Adaptive stratified importance sampling: hybridization of extrapolation and importance sampling Monte Carlo methods for estimation of wind turbine extreme loads". We include your comments for reference, followed by point-by-point replies.

**Review: Adaptive stratified importance sampling: hybridization of extrapolation and importance sampling Monte Carlo methods for estimation of wind turbine extreme loads**

This paper proposes an approach for estimating wind turbine extreme loads by integrating the importance sampling method and extrapolation technique. In the past, extrapolation has been widely used for estimating extreme loads, but existing extrapolation methods do not consider the unequal importance of different wind speed bins on the estimation accuracy. Built on the importance sampling, the authors propose to use unequal sample sizes in different bins according to their importance in terms of variance minimization. The proposed approach is interesting in that it improves existing extrapolation methods through the adaptive sampling procedure. I have some comments to improve the methodology, notations and presentations.

1. First of all, notations are confusing. The quantity to be estimated (e.g., $P(Y < l)$) is different from its estimator. For the estimator, "hat" is conventionally used above the quantify (e.g., $\hat{P}(Y < l)$).

2. For clarity, I suggest that the estimator to estimate POE needs to be explicitly defined in a mathematical form. Here, the authors use the iterative method, so the estimator needs to incorporate the iteration index.

3. Even though the summary of the procedure is given in Section 3.2, this procedure does not contain the detailed information to implement the approach. For example, in Step 4, how to update the empirical estimates and extrapolation estimates? For extrapolation, will the data from the last iteration be used, or will data obtained from all iterations used?

4. The goal of this paper is to develop methods that make unbiased estimates while minimizing variance. But extrapolation cannot guarantee the unbiased estimation, as it approximates the conditional density with statistical models (3 parameter Weibull distribution in this paper). Even though the statistical models can be improved throughout iterations, the models are still surrogate models. Therefore, it should be discussed/justified how the proposed approach can provide a unbiased estimation.

5. It is not clear how $g(x_i)$ becomes $\frac{N_i}{N_{tot}}\frac{1}{\Delta x_i}$. Because $g(x_i)$ is a density, $\int g(x)dx$ should be 1, which is not in the proposed form. The authors might consider $g(x_i) \propto \frac{N_i}{N_{tot}}\frac{1}{\Delta x_i}$.

6. The most important question is what is the benefit of the proposed approach over the stratified sampling and importance sampling. Stratified sampling provides the closed-form of optimal $N_i$'s. So, it is not clear why one needs to use the stochastic optimization approach combined with the importance sampling in Sec. 3.2. Also, what is the benefit of the proposed approach over the method proposed in Choe et al. (2016) (or their prior study) that uses importance sampling only?

7. In the implementation results, the relative standard deviations are presented. To show whether the estimates are unbiased or not, the POE estimates (or extreme load estimates) from the approach should be compared with those from crude Monte Carlo (e.g., authors may compare the estimates with those in the following paper: Barone, M., Paquette, J., Resor, B., and Manuel, L., 2012, "Decades of Wind Turbine Load Simulation," AIAA Paper No. 2012–1288.)

Replies:

1. Thank you for pointing out the convention, and I am sorry if the current notation is not clear. I am aware of the convention to use "hat". However, I do notice, even in well-regarded statistical texts (e.g. *Monte Carlo Statistical Methods*, Robert and Casella), a certain lack of consistency, that seems to derive largely from the difference between statements such as 1) "The estimate of interest is $X = E_f[Y] = \int Y(x)f(x)dx \sim \sum_i Y(x_i)f(x_i)$", where we have switched from exact to estimate in the middle of the series of steps, where it would be wrong to start with "$\hat{X} = ...$", and 2) "The estimate of $X = E_f[Y] = \int Y(x)f(x)dx$ is $\hat{X} = \sum_i Y(x_i)f(x_i)$". In an effort to be consistent, and a personal preference for less rather than more symbols, and for using statements such as 1) above to keep the equations "flowing", I have simply omitted "hats". I have 1) gone through the text and tried to be sure we are clearly indicating when something is an estimate versus an exact quantity and 2) rephrased the initial description of "exceedance probability" to include the "hat".

2. We have added language referring to the explicit equations one would use to form the empirical estimate at each iterations. I agree that we could also label the estimates at each iteration with an iteration index, e.g., something like $P^k(Y < l)$, where $k$ is the ASIS iteration. I think this, like the hats, represents the introduction of more symbols than are necessary for the level of detail we are currently targeting. We plan to write a shorter "recipe-like" companion paper in which the algorithm is very clearly stated in terms that the reader could readily implement.

3. This is admittedly just a sketch of the algorithm. To answer your specific question, the data used at each iteration is cumulative. We have added language to that effect to the outline. This paper is about opening new ground (specifically, seeing how an initial bin-based conception can be transformed into stochastic search for the optimal importance density), and the details of the tuned version of this algorithm are still being worked out. As stated above, at this point we would intend to write another paper in which is much closer to a simple recipe for how to implement the recommended algorithm.

4. Perhaps we have not been clear. The ASIS method is a rigorously justified unbiased importance sampling estimate. As you point out, the extrapolative estimates are emphatically not unbiased estimates. We will make sure this is clearly stated in the text.

5. We claim $g(x)$ is in fact normalized:

$$
\begin{aligned}
\int g(x)dx &= \sum_{i=1}^{N_{bins}} \int_{x_i}^{x_{i+1}} g(x_i)dx \\
&= \sum_{i=1}^{N_{bins}} \int_{x_i}^{x_{i+1}} \frac{N_i}{N} \frac{1}{\Delta x} dx \\
&= \sum_{i=1}^{N_{bins}} \frac{N_i}{N} \frac{1}{\Delta x} \Delta x \\
&= \sum_{i=1}^{N_{bins}} \frac{N_i}{N} \\
&= 1.
\end{aligned}
$$

We now explain this in the text.

6. In Choe et al (2016 and prior work), one will notice that the closed form optimal $N_i$ contains a term, $s(x) = P(Y > l | X = x)$, that is not known in practice, and in fact is the quantity that they are trying to estimate. To overcome this difficulty, Choe et al run a certain (in there case I believe 250) number of simulation, from which they build a surrogate model of this function $s(x)$, only after which they can use their optimal $N_i$ values. Also, the optimal $N_i$ depends on what load value $l$ they are interested in. Importance sampling is a way of estimating with respect to one distribution by sampling from another. But adaptive importance sampling is made difficult by the need to keep track of the exact importance distribution even while it is changing. The use of a bin-based ("stratified") formulation allows for precisely defining the importance distribution at each iteration. This amounts to a form of "parameterization" of the importance distribution. Our main contribution is not as much in the optimization of the bin distribution but more in deriving a formulation that explicit converts any bin distribution into an unbiased IS estimate.

7. We have done this in our earlier paper: Graf, P., Damiani, R., Dykes, K., and Jonkman, J.: Advances in the Assessment of Wind Turbine Operating Extreme Loads via More Efficient Calculation Approaches, in: AIAA SciTech 2017.

Again, thank you very much for the careful and thoughtful review of our paper. I hope I have addressed your concerns. Your comments helped improve the paper considerably.

Sincerely,

Peter Graf, on behalf of the authors.

Dear Dr. Dimitrov ("referee 3"),

Thank you very much for the thoughtful and thorough review of our paper, "Adaptive stratified importance sampling: hybridization of extrapolation and importance sampling Monte Carlo methods for estimation of wind turbine extreme loads". I know it takes a lot of time to provide such feedback. I have really learned from your comments. In particular, connecting work on wind turbine extreme loads to the larger body of literature on structural reliability is important, and I would aspire to do a better job of that going forward. What follows are your specific comments, followed by my replies.

*1) As the authors say on Page 2, line 9: "Monte Carlo methods: exceedance probabilities are written as expectations of indicator functions". This is in fact a passing definition for structural reliability problems, regardless of the method used for the integration (the computation of the expected value). We can therefore make a parallel between the statistical extrapolation and the structural reliability analysis problems. This means we can take inspiration from literature where e.g. adaptive importance sampling is used as a tool for reliability analysis. Some examples include [1], [2] where search-based (adaptive) importance sampling is formulated and also [3] where it is applied to a dynamical system.*

Certainly, our work is in the category of structural reliability, so these references are relevant. I want to be clear that the ASIS method is NOT extrapolation. The common ground is the bin-based formulation that allows for samples that would "traditionally" be used for extrapolation to also be used for importance sampling, which are non-extrapolative unbiased estimates. Thank you for providing these excellent background references on adaptive importance sampling. I am including these in the background material.

*2) The current results section mainly shows the behaviour of the ASIS algorithm, but it is impossible to judge how it will compare to standard extrapolation procedures using the same number of FAST simulations. The algorithm suggested by the authors is essentially an approach for "guided" sampling from the joint distribution of environmental conditions. It would be really interesting to see a comparison with e.g. direct sampling from the long-term distribution using a pseudo Monte-Carlo simulation with low-discrepancy series, followed by extrapolation directly from the empirical CDF of the long-term load distribution.*

Comparing ASIS to "standard extrapolation" is a bit of an apples-to-oranges comparison. Extrapolation relies on assuming and fitting an extreme value distribution, after which loads corresponding to arbitrarily small exceedance probabilities can be estimated. Whereas, ASIS is specifically formulated to provide unbiased (Monte Carlo) estimates of the exceedance probabilities, where the price we pay for the lack of assumption of any particular extreme value distribution is that we have to "earn" the low exceedance probabilities by perhaps more samples. I think your suggestion ("*pseudo Monte-Carlo simulation with low-discrepancy series, followed by extrapolation directly from the empirical CDF*") represents the best "competition" to ASIS (or, in general, any form of adaptive sampling) from a non-adaptive approach, so indeed it would be interesting to try. However, this experiment is beyond the scope of the *present* paper, where the intent was largely to show how the bin-based approaches that are typically

associated with extrapolation could be adapted for use in importance sampling, rather than to claim we have found the strictly best approach.

3) *Statistical extrapolation is a very academic exercise. In the last years the wind energy scientific society has published multiple papers which present attempts at improving the efficiency and reducing the uncertainty in load extrapolation. But the majority of these solutions represent highly complex algorithms with many parameters to tune and with tricky implementation which can easily be done in a wrong way. My general concern with statistical extrapolation is that this complexity hinders the adoption of new research in industry as it requires a significant level of very specific expertise in order to achieve a correct and robust implementation. As a confirmation to this observation, the coming IEC 61400-1 ed.4 standard actually suggests methods for doing extrapolation with even lower complexity (and most probably higher uncertainty) than what was suggested in ed.3 of the same standard. This should not be viewed as a criticism to the present paper or its authors – but in my opinion the focus of future research in statistical extrapolation should not only be towards improving accuracy or efficiency, but also towards achieving robust and easy to implement solutions. This is something the authors may want to give a thought to and eventually discuss in their paper.*

This is a great point.  However, 1) the paper *is* an academic exercise, 2) the codes that simulate the dynamics of the turbine response are also very complex.  Perhaps what is needed is not a simplified procedure but a set of open source community codes that perform the statistical analysis.  That said, although this paper is reaching to develop a new method, resources permitting, we intend to write a companion paper that distills ASIS to a simple, easily implementable, "recipe".

4) *Page 5, line 24: The authors use 10 peaks per 10-minute simulation. However, they do not seem to take any measures for ensuring statistical independence between the peaks (or there is no description of any measures they have taken). Thus, there is a very high chance that some of the events they have identified as peaks are correlated (e.g. they have small time separation). The result is most likely a (slight) over-prediction of the exceedance probabilities. A simple and straightforward measure for reducing dependency could be enforcing some minimum time separation between successive 1-minute peaks. Typically time separation equivalent to the time for 1-2 rotor revolutions should be sufficient to eliminate dependence almost entirely.*

In an effort to prevent the analysis from being more complicated than it already is, we have made the assumption that peaks from each 1 minute segment are independent.  We would argue that the error in the resulting estimates resulting from peaks that are less than 1-2 rotations apart but happen to fall on either side of a 1 minute division is not significant compared to the variation due to 1) stochastic turbine response 2) lack of full convergence of the Monte Carlo estimates.

5) *Figure 1: The amount of variability in the peaks is also a consequence of the authors' choice to use one-minute peaks. Drawing the same plot for the 120 ten-minute peaks per bin would show lower variation.*

This is a fair point, given that 9 out of our 10 one-minute peaks will be smaller than the ten-minute peak.  But we argue (anecdotally) that the reduction in variability using only ten-minute peaks is much smaller than the variability itself.  There is a limit, we admit; for example, using the peaks every five-seconds would be misleading (there would be lots of small peaks), but one-minute peaks are defensible, and helps us develop method faster because we get 10 times as much data to analyze.

*6) Section 2.2: Some good additional references to consider in this section are [4] which presents an excellent method for accounting for independence between peaks, [5] which is an extensive parametric study on extrapolation considering, among other things, the effect of number of peaks on extrapolation accuracy, and [6] where the seed-to-seed variability is addressed explicitly, though with focus on fatigue loads.*
These are excellent and appreciated suggestions.  The literature on peak independence, numbers of peaks, thresholds for peak-over-threshold, appropriateness of difference statistical distributions, etc., is indeed extensive.  Thank you for the suggestions, we have added the references suggested.

*7) Section 2.4: I think this section can be removed or shortened significantly without reducing the quality of the paper. To me, what the authors try to define in the section is that the extrapolation per wind speed represents a set of conditional approximations which can best be described by a Rosenblatt transformation [7]. The IFORM method is related to the Rosenblatt transformation as the Rosenblatt transformation is a convenient way to draw the IFORM contours. Further, the IFORM method is meant to be applied to static quantities as e.g. 10-minute statistics of environmental conditions. The inventor of the method (S. Winterstein) has also worked on re-formulating the IFORM for dynamic problems as the one considered by the authors of the present paper [8]. Another relevant method is the tail-equivalent linearization method (TELM) [9], which employs FORM for approximation of the extreme response of dynamical systems. In general my feeling is that discussing FORM or IFORM in the current version of this paper is a distraction from the main scope – but if it is necessary to retain it, the TELM method could be a possible bridge.*

We have explicitly called out this section as "optional", as you are not the only reviewer to think it does not fit in that well.  The point was about response variability: if the same environmental conditions can result in very different response (due to "random seeds" used, e.g., to generate turbulent inflow), then there is very little that can be done to reduce variance of estimates other than run large numbers of simulations.  In our discussion, this is related to IFORM through the "environmental contour" method.  Thank you for the suggestion regarding TELM, that is an interesting approach.  Undoubtedly many interesting connections could be made between the question of "number of peaks to use" which is a proxy for "threshold" in peak detection, and the TELM method, because, like TELM, we are using only the "tail" data to fit our extrapolation distribution.  This field, we feel, has the potential for much of this sort of unification of methods previously assumed to be disparate (this is exactly what we have done in in building a "bridge"

from bin-based extrapolation to importance sampling), but it is beyond the scope of the present paper.

*8) Page 11, lines 15-25: As the authors state, this algorithm will converge to a local optimum, and measures should be taken to ensure that the global optimum is found. We could again make the parallel to structural reliability, where multiple failure modes lead to a system reliability problem with multiple design points ("local minima") and the failure domain may be non-convex. When employing (adaptive) importance sampling, the system reliability problem is normally approached by using multiple "seeds" of importance sampling densities which are initially placed in the different parts of the variable domain. Melchers with multiple design points ("local minima") and the failure domain may be non-convex. When employing (adaptive) importance sampling, the system reliability problem is normally approached by using multiple "seeds" of importance sampling densities which are initially placed in the different parts of the variable domain. The authors may want to consider such an approach for their problem – it can potentially also help with the issue of the largest variance being at different wind speeds for different load channels.*

The non-convexity of the search for the optimal bin-distribution is different from (but probably complicated by) the interest in making estimates for different load channels at the same time. Our interest in this paper has been more toward formulating the extreme loads estimation problem as stochastic/global search, not necessarily in providing the optimal solution algorithm. Therefore, we have simply adopted the common strategy for global search (e.g., simulated annealing) of allocating a certain percentage of the search steps to "exploration" rather than "exploitation" (gradient descent). It is certainly a good and intuitive idea to build the overall IS distribution from individual distributions targeted toward different load channels.

*9) Figure 2: A good reference for the effect of number of peaks (time series) on the extrapolation is [5], where this is investigated for several types of extrapolation methods, distribution fits, and with up to 1000 minutes of time series per extrapolation. Another relevant study is the one by Zwick & Muskulus [6] where they thoroughly investigate the seed-to-seed variability problem, though without considering extrapolation.*

Thanks for these suggestions. We refer to these now at appropriate points in the text.

*Technical comments:*
*10) Page 7, line 28: Y*f] is probably a typo. Did the authors mean [Y*f]?*

Indeed, this is a typo, thank you.

Thank you very much for taking the time to prepare such a thoughtful review of our paper, in particular for providing such a wealth of references for us to learn from. They have certainly allowed us to improve the level of discussion in the paper. I apologize for not being able to pursue all your suggestions thoroughly in the present paper, but they will be very helpful going forward.

Sincerely,

Peter Graf, on behalf of the authors.